# Antagonistic regulation by insulin-like peptide and activin ensures the elaboration of appropriate dendritic field sizes of amacrine neurons

Jiangnan Luo[1], Chun-Yuan Ting[1], Yan Li[2], Philip McQueen[3], Tzu-Yang Lin[2], Chao-Ping Hsu[4,5], Chi-Hon Lee[1,2]*

[1]Section on Neuronal Connectivity, Eunice Kennedy Shriver National Institute of Child Health and Human Development, National Institutes of Health, Bethesda, United States; [2]Institute of Cellular and Organismic Biology, Academia Sinica, Taipei, Taiwan, Republic of China; [3]Mathematical and Statistical Computing Laboratory, Center for Information Technology, National Institutes of Health, Bethesda, United States; [4]Institute of Chemistry, Academia Sinica, Taipei, Taiwan, Republic of China; [5]Genome and Systems Biology Degree Program, National Taiwan University and Academia Sinica, Taipei, Taiwan, Republic of China

*For correspondence:
leechih@gate.sinica.edu.tw

Competing interests: The authors declare that no competing interests exist.

**Abstract** Establishing appropriate sizes and shapes of dendritic arbors is critical for proper wiring of the central nervous system. Here we report that Insulin-like Peptide 2 (DILP2) locally activates transiently expressed insulin receptors in the central dendrites of *Drosophila* Dm8 amacrine neurons to positively regulate dendritic field elaboration. We found DILP2 was expressed in L5 lamina neurons, which have axonal terminals abutting Dm8 dendrites. Proper Dm8 dendrite morphogenesis and synapse formation required insulin signaling through TOR (target of rapamycin) and SREBP (sterol regulatory element-binding protein), acting in parallel with previously identified negative regulation by Activin signaling to provide robust control of Dm8 dendrite elaboration. A simulation of dendritic growth revealed trade-offs between dendritic field size and robustness when branching and terminating kinetic parameters were constant, but dynamic modulation of the parameters could mitigate these trade-offs. We suggest that antagonistic DILP2 and Activin signals from different afferents appropriately size Dm8 dendritic fields.

## Introduction

The establishment of complex yet stereotyped neuronal dendritic arbors is a key determinant for proper wiring of the central nervous system. The size and complexity of the dendritic trees govern the number of inputs a neuron can sample, while the location of dendritic arbors determines the types of neuronal inputs the neuron receives (*Luo et al., 2016*; *Lefebvre et al., 2015*). Both intrinsic and extrinsic mechanisms regulate various aspects of neuronal dendritic development, such as the elongation and targeting of dendrites (*Dong et al., 2015*) or restriction of dendritic expansion by self-avoidance and afferent-derived signals (*Jan and Jan, 2010*; *Ting et al., 2014*). Developing dendrites exist in complex dynamic microenvironments and exhibit phase-specific characteristics in vivo (*Dailey and Smith, 1996*; *Wu et al., 1999*). Cell–cell interaction events punctuate the stages of dendrite outgrowth (*Jan and Jan, 2010*; *Lefebvre et al., 2015*), with dendrites responding to afferents (*McAllister et al., 1995*; *Joo et al., 2014*) and neighboring dendrites (*Yamagata et al., 2002*; *Lefebvre et al., 2012*; *Matthews et al., 2007*). Cell surface receptors, such as Cadherins and Dscams, mediate adhesion or repulsion to promote dendritic branching complexity and self-

avoidance (*Togashi et al., 2002*; *Tanabe et al., 2006*; *Hughes et al., 2007*; *Matthews et al., 2007*; *Soba et al., 2007*; *Deans et al., 2011*; *Tadros et al., 2016*). Secreted factors, such as BDNF, semaphorins, TGF-β ligands and Wnt-family proteins, act on target cells via cognate receptors to stimulate specific intracellular signaling pathways that regulate dendritic branching pattern and complexity (*McAllister et al., 1995*; *Horch and Katz, 2002*; *Kirszenblat et al., 2011*; *Shelly et al., 2011*; *Ting et al., 2014*; *Liao et al., 2018*). While an increasing number of signaling molecules have been identified as key regulators of dendrite development, it remains poorly understood how multiple signaling pathways are coordinated to meet unique developmental needs.

To study the mechanisms of dendrite development in the central nervous system, we utilized *Drosophila* optic lobe neurons as a model (*Ting et al., 2014*). The *Drosophila* optic lobe shares many common features with vertebrate retinae, including columnar and laminar organization and diverse neuronal types (*Sanes and Zipursky, 2010*). The largest optic neuropil, the medulla, is organized in columns and layers (M1-M10), and it receives direct inputs from narrow-spectrum R7 and R8 photoreceptors at the M6 and M3 layers, respectively. Indirect inputs to this tissue come from broad-spectrum R1-R6 photoreceptors via lamina neurons L1-L5 at multiple layers (*Figure 1A*; *Fischbach and Dittrich, 1989*). The regularly patterned axons of R7, R8 and lamina neurons form a lattice-like structure, thereby establishing retinotopic maps in the medulla.

The medulla has over sixty types of medulla neurons, each of which extends dendritic arbors into distinct layers with type-specific field sizes (*Fischbach and Dittrich, 1989*; *Takemura et al., 2013*; *Figure 1A*). Among these neurons, Dm8 amacrine neurons have been extensively studied for their function in mediating spectral preference behaviors (*Gao et al., 2008*; *Karuppudurai et al., 2014*). During development, Dm8 neurons extend dendrites into the medulla M6 layer at the early pupal stage, and the dendritic tree gradually expands to its full size, which receives synaptic inputs from approximately 14 R7 photoreceptors (*Ting et al., 2014*). The R7-derived morphogen, Activin, acts at short range to restrict Dm8 dendritic field expansion, and disruption of Activin signaling by removing its receptor, Baboon, or the downstream target, Smad2, aberrantly expands Dm8 dendritic fields (*Ting et al., 2014*). On the other hand, the mechanisms that promote Dm8 dendritic growth are not known; thus, the regulatory mechanism for dynamic expansion of Dm8 dendritic fields remains elusive.

The evolutionarily conserved insulin/insulin-like growth factor (IGF) signaling pathway regulates many aspects of neuronal development and function (*Chiu and Cline, 2010*). In *Drosophila*, eight insulin-like peptides (DILP1-8), which are homologous to vertebrate insulin and IGF-I, regulate a wide range of cellular processes, including cellular growth, metabolism, fertility and longevity (*Grönke et al., 2010*; *Colombani et al., 2012*; *Garelli et al., 2012*). DILP2, 3 and 5 are secreted from the median insulin-producing cells, while DILP6 is secreted from the larval fat body, yet all of these DILPs are thought to function mostly as endocrine hormones (*Ikeya et al., 2002*; *Slaidina et al., 2009*). Interestingly, glia-derived DILPs appear to function locally as well, reactivating neuronal stem cells from quiescence and inducing neuronal differentiation during development (*Chell and Brand, 2010*; *Fernandes et al., 2017*). Activated insulin receptor is capable of stimulating several signaling pathways, including the Ras/MAPK and PI3K pathways, which are primarily responsible for the respective roles of the insulin receptor in cell proliferation and metabolism (*Siddle, 2012*). In addition, the *Drosophila* insulin receptor (InR) functions in photoreceptor axons as a guidance receptor, which signals through the adaptor protein Dock/Nck and the p21-activated kinase to regulate the actin-cytoskeleton (*Song et al., 2003*). Insulin signaling regulates various aspects of dendritic development, including morphogenesis, plasticity and regeneration (*Chiu et al., 2008*; *Agostinone et al., 2018*). However, the source of insulin/insulin-like peptide and the signaling mechanisms involved remain unclear.

Here we report that insulin-like peptide DILP2 from developing L5 lamina neurons positively regulates dendrite morphogenesis of Dm8 neurons. Our use of a split-GFP-tag technique then revealed that during a critical time window, the DILP2 signal is received by endogenous insulin receptors concentrated in the center region of Dm8 dendrites. Activated insulin receptor signals through the PI3K/TORC1 pathway via SREBP (sterol regulatory element binding protein; a transcription factor that acts as a central regulator of lipogenic gene expression) to promote Dm8 dendritic growth. We further showed that this positive regulation by insulin signaling operates in parallel with the known negative regulation by Activin signaling; together these signaling pathways constitute a 'push-pull' mechanism that robustly controls Dm8 dendritic field elaboration.

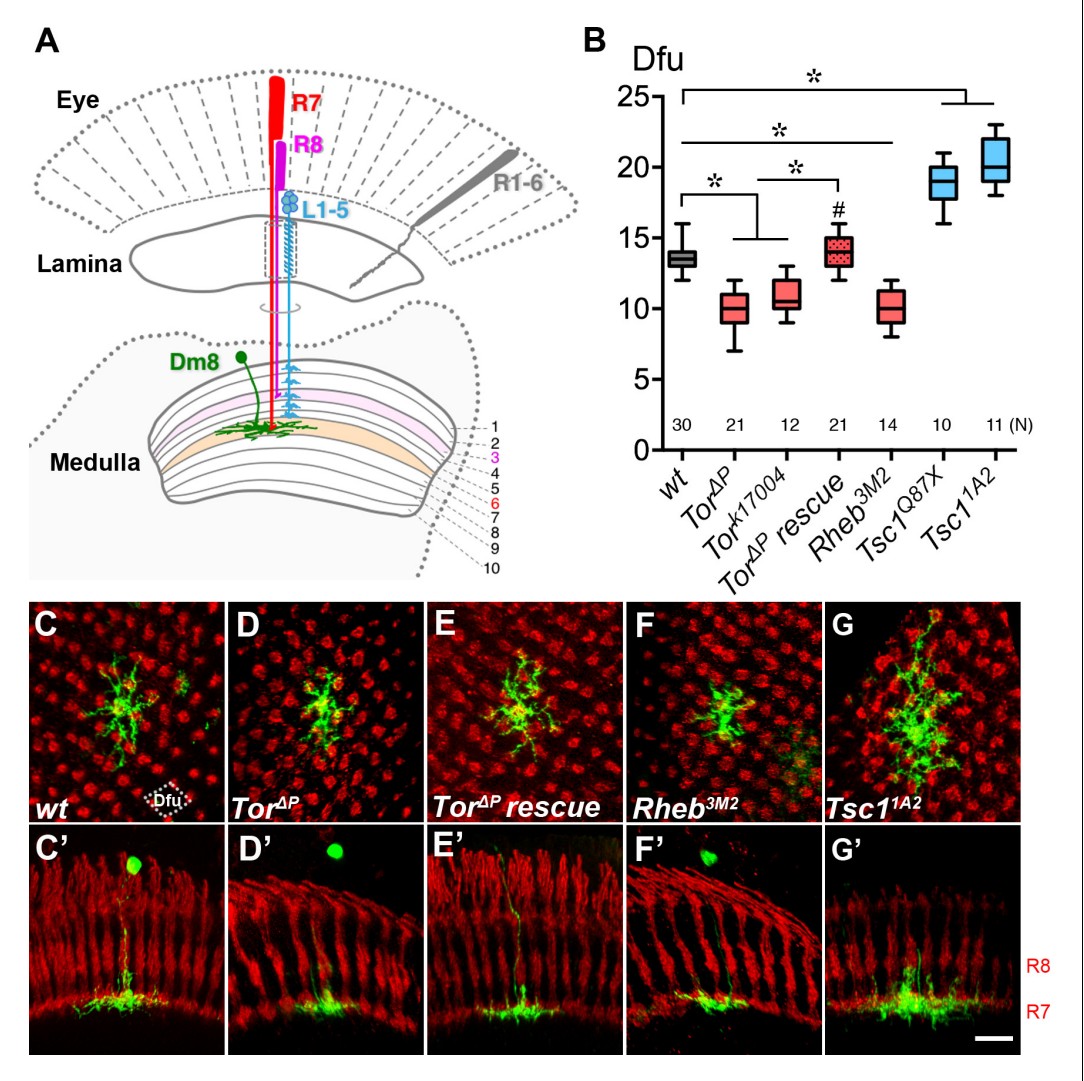

**Figure 1.** Cell-autonomous TOR signaling positively regulates the dendritic field sizes of Dm8 amacrine neurons. (**A**) A schematic illustration of the *Drosophila* peripheral visual system, including the eye and two optic neuropils, the lamina and the medulla. The chromatic photoreceptors R7 (red) and R8 (magenta) extend axons into the M6 and M3 layers of the medulla neuropil, respectively, while the broad-spectrum R1-6 photoreceptors (grey) terminate their axons in the lamina. The wide-field amacrine neuron Dm8 elaborate their dendrites in the M6 layer to receive inputs from about 14 R7s. The axons of the lamina neurons L1-5 (cyan) terminate in various M1-M5 layers. (**B**) Box plot showing the dendritic field sizes of wild-type and various mutant Dm8 neurons. For *Tor* and *Tsc1*, two different null mutant alleles were tested. The Dm8 dendritic field sizes were quantified by counting the number of encompassed dendritic field units (dfu). One dfu was defined as a rhombus area with four neighboring photoreceptor terminals as vertexes (indicated by dotted lines in (**C**)). Expression of wild-type Tor significantly rescued *Tor* mutant Dm8 dendritic phenotypes. *p<0.05; unpaired Student's t test. N; the number of cells scored for each genotype. (**C–G**) Single wild-type (**C,C'**), *Tor* mutant (**D,D',E,E'**), *Rheb* mutant (**F,F'**) and *Tsc1* mutant (**G, G'**) in adults. Dm8 neurons were generated and labeled with the mCD8GFP marker (green) using the hs-flp/ MARCM system. The photoreceptor axons, labeled with anti-24B10 (red), were used as landmarks. (**C–G**) show proximal-distal views while (**C'–G'**) show the corresponding dorsal-ventral views. As compared to wild-type neurons (**C,C'**), *Tor* (**D,D'**) and *Rheb* (**F,F'**) mutant Dm8 dendritic fields were smaller, covering fewer dendritic field units (dfu), while *Tsc1* (**G,G'**) mutant Dm8 dendritic fields were larger. The dendritic field size of *Tor* mutant Dm8 neurons was restored by overexpressing a wild-type form of *Tor* during development (**E,E'**). The layer-specific targeting of Dm8 dendrites was unaffected in all mutants examined (**C'–G'**). Scale bar (shown in G' for C-G'), 5 μm. The online version of this article includes the following figure supplement(s) for figure 1:

**Figure supplement 1.** Cell fate is unaffected in *Tor* and *Pten* mutant Dm8 neurons.

## Results

### Canonical TOR signaling positively regulates Dm8 dendritic field size

To identify genes required for proper dendritic patterning of Dm8 neurons, we screened a collection of known mutants for Dm8 dendritic morphological phenotypes. Because our previous study suggested cell-cell communication is involved in Dm8 dendritic patterning, we targeted cell surface receptors and cytoplasmic signaling molecules and screened available mutants using mosaic analyses. This candidate approach revealed that a *Tor* (target of rapamycin) null mutation (*Tor$^{\Delta P}$*) (*Zhang et al., 2000*) affects Dm8 dendritic patterning (*Figure 1B*). TOR, a serine/threonine kinase, is known to regulate a broad range of cellular processes, such as lipid synthesis, translation, autophagy and cytoskeletal changes (*Wullschleger et al., 2006*; *Betz and Hall, 2013*). Using the MARCM (mosaic analysis with a repressible marker) technique, we then generated wild-type and *Tor* mutant Dm8 clones in an otherwise heterozygous background and examined their dendritic morphologies in adult animals (*Lee and Luo, 1999*; *Ting et al., 2014*). In contrast to the wild-type Dm8 dendrites, which covered approximately 14 R7 terminals in the medulla M6 layer (*Ting et al., 2014*; *Figure 1C and C'*), the *Tor$^{\Delta P}$* mutant Dm8 dendritic fields appeared to be smaller (*Figure 1D and D'*). However, the other dendritic morphological attributes, including specific targeting to the medulla M6 layer, were indistinguishable from wild type (*Figure 1C–D'*). We did not notice drastic difference in the frequencies of wild-type and *Tor* mutant clones but we cannot rule the possibility that very few mutant clones died and went undetected because of the sampling nature of the mosaic technique. We quantified the sizes of Dm8 dendritic fields using two metrics: the number of R7 terminals (nc) contacted by Dm8 dendrites, and the number of dendritic field units (dfu, defined as a rhombus area with four neighboring R7 terminals as vertexes, *Figure 1C*) occupied by Dm8 dendritic arbors (*Ting et al., 2014*). We found that the *Tor$^{\Delta P}$* Dm8 neurons had a reduced dendritic field (10.6 ± 0.3 nc; 9.9 ± 0.3 dfu [n = 21]) compared to wild-type neurons (14.5 ± 0.2 nc; 13.5 ± 0.2 dfu [n = 30]) (*Figure 1B*). The dendritic caliber might be affected in mutant Dm8s but we were not able to assess this phenotype due to the resolution limitation of the light microscopy. We further examined mutant Dm8 clones of another *Tor* mutant allele *Tor$^{k17004}$* and found that this mutation produced a similar dendritic field reduction phenotype (11.3 ± 0.3 nc; 10.8 ± 0.3 dfu [n = 12], *Figure 1B*). Next, we carried out a transgene rescue experiment by restoring TOR in *Tor* mutant Dm8 MARCM cells. The transgene-mediated expression of a wild-type form of TOR (*Tor$^{WT}$*) significantly rescued the *Tor$^{\Delta P}$* dendritic phenotype in Dm8 neurons (14.8 ± 0.3 nc; 14.0 ± 0.2 dfu [n = 21] (*Figure 1E*). Together, these results indicate that TOR is required cell-autonomously in Dm8 neurons for expanding proper sizes of dendritic fields. To determine if the observed dendritic phenotypes reflected an altered cell fate, we examined the expression of cell-type-specific transcription factors in wild-type and mutant Dm8 neurons. We found that *Tor* mutant Dm8s and wild-type controls all expressed Dachshund (Dac) but not Drifter (Dfr), Twin of Eyeless (Toy), Dichaete (D), or Eyeless (Ey) (*Figure 1—figure supplement 1A–M'*), suggesting that the normal cell fate was retained. The cell body sizes of *Tor* mutant Dm8 neurons were comparable to wild type while those of *Pten* (a negative regulator of PI3K/AKT signaling) mutants are larger (*Figure 1—figure supplement 1E*). Taken together, these results suggest that TOR signaling specifically regulates Dm8 dendritic field size and is cell-autonomously required for Dm8 dendrite development.

We next tested whether *Rheb (Ras homolog enriched in the brain)* and *Tsc1(Tuberous sclerosis complex 1),* two immediate upstream regulators of TOR, regulate Dm8 dendritic field elaboration. In flies and other organisms, Tsc1-Tsc2 inhibits Rheb, which in turn activates TOR activity (*Saucedo et al., 2003*). As expected, *Tsc1* mutant Dm8 neurons exhibited drastically expanded dendritic fields (20.4 ± 0.5 dfu for *Tsc1$^{1A2}$* [n = 11]; 18.7 ± 0.5 dfu for *Tsc1$^{Q87X}$* [n = 10]) (*Figure 1G*), while *Rheb$^{3M2}$* mutant Dm8 neurons displayed the opposite phenotype (10.2 ± 0.4 dfu [n = 14]) (*Figure 1F*). These findings suggest that Tsc1 and Rheb regulate Dm8 dendritic field size, likely by coupling TOR signaling to upstream signaling.

### Insulin signaling regulates Dm8 dendritic field sizes

Previous studies have shown that the Tsc1/Rheb/Tor pathway can be activated by insulin/IGF signaling via the adaptor protein IRS (Chico), PI3 kinase (PI3K) and AKT kinase (*Figure 2O*) (reviewed in *Grewal, 2009*; *Clancy et al., 2001*; *Saucedo et al., 2003*). Therefore, we next examined two

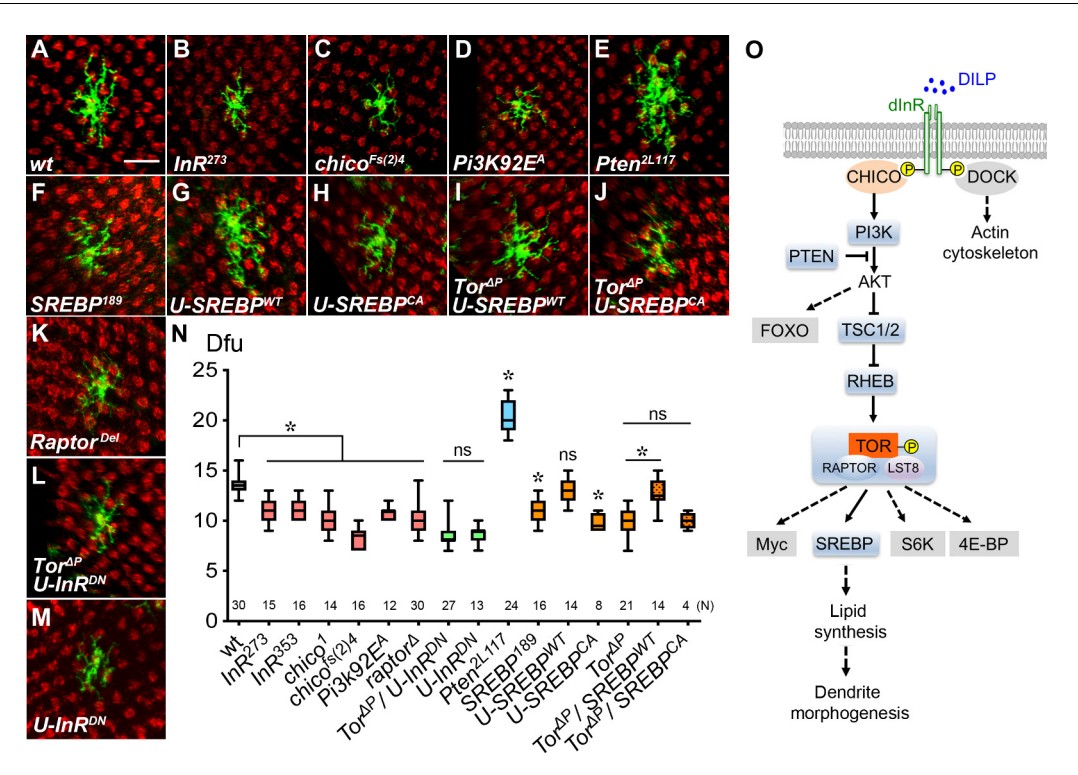

**Figure 2.** Insulin/Tor signaling positively regulates Dm8 dendrite development through SREBP. (A–F) Insulin/Tor signaling and the downstream regulator SREBP are cell-autonomously required for normal Dm8 dendritic size and patterning. Single wild-type (A), $InR^{273}$ (B), $chico^{Fs(2)4}$ (C), $Pi3K92E^A$ (D), $Pten^{2L117}$ (E), $SREBP^{189}$ (F) and $Raptor^{Del}$ (K) were generated and labeled with mCD8GFP marker (green) using the hs-flp/MARCM system and examined in adults. The photoreceptor axons, labeled with anti-24B10 (red), were used as landmarks. As compared to wild type (A), mutation of *InR* (B), *chico* (C), *Pi3K* (D), *SREBP* (F) and *Raptor* (K) significantly decreased Dm8 dendritic field sizes, while mutation of *Pten* (E) enlarged Dm8 dendritic field size. Scale bar, 10 μm. (G–H) Using DIP-γ-Gal4 to generate Dm8 MARCM clones and simultaneously overexpressing a wild-type form of *SREBP* ($SREBP^{WT}$) (G) resulted in no changes of Dm8 dendritic field size, while overexpressing a constitutively active form of *SREBP* ($SREBP^{CA}$) decreased Dm8 dendritic field size (H). (I–J) Overexpression of $SREBP^{WT}$ (I), but not $SREBP^{CA}$ (J), rescued the dendritic phenotypes of *Tor* mutant Dm8 neurons. Overexpressing a dominant negative form of InR (U-InR^DN) in *Tor* mutant (L) or otherwise wild-type (M) Dm8s decreased dendritic field size. (N) Box plot showing dendritic field sizes of wild-type and various mutant Dm8 neurons. *p<0.05; ns, not significant; unpaired Student's t-test. N; the number of cells scored for each genotype. (O) A schematic drawing depicts the core components in the insulin/Tor signaling pathways that regulate Dm8 dendritic field size. The tested components shown with grey shading are not required for Dm8 dendritic field size regulation. Dashed lines indicate probable but unconfirmed cascades.

The online version of this article includes the following figure supplement(s) for figure 2:

**Figure supplement 1.** Genetic mutations of several components in the insulin/Tor signaling cascade do not alter Dm8 dendritic field size.

hypomorphic alleles of *InR* and found that *InR* mutant Dm8 neurons, like *Tor* mutants, had smaller dendritic fields than wild types (10.9 ± 0.3 dfu [n = 15] for $InR^{273}$; 11.1 ± 0.3 dfu [n = 16] for $InR^{353}$, **Figure 2B and N**), suggesting insulin signaling is indispensable for proper Dm8 dendrite development. Expressing a dominant negative form of insulin receptor (InR^DN) in otherwise wild-type Dm8s led to small dendritic field sizes (8.5 ± 0.2 dfu [n = 13] for *U-* InR^DN, **Figure 2M,N**). Similarly, Dm8 mutants devoid of the adaptor protein Chico or PI3K had small dendritic fields (8.4 ± 0.3 dfu [n = 16] for $chico^{fs(2)4}$; 10.3 ± 0.4 dfu [n = 14] for $chico^1$, 10.8 ± 0.2 dfu [n = 12] for $PI3K^{Df(3R)}$; **Figure 2C, D and N**). Conversely, Dm8 mutants devoid of PTEN, a negative regulator of PI3K/AKT signaling, exhibited larger dendritic fields (20.4 ± 0.3 dfu [n = 24] for $Pten^{2L117}$, **Figure 2E and N**) and increased cell-body sizes (**Figure 1—figure supplement 1E**) compared to wild-type controls. In

contrast, mutant Dm8 neurons devoid of the adaptor protein, Dock, or the transcription factor, FOXO (two branching pathways downstream of InR and AKT, respectively) had wild-type-like dendritic fields (13.7 ± 0.3 dfu [n = 11] for *dock* [k13421]; Foxo[Δ94], 13.9 ± 0.3 dfu [n = 15]; *Figure 2—figure supplement 1B and C*). To further examine the relationship between TOR and InR in Dm8 dendrite development, we tested genetic interaction and found that overexpressing the dominant negative form of InR (InR[DN]) in *Tor* mutant Dm8 neurons led to InR[DN]-like dendritic field sizes (8.5 ± 0.2 dfu [n = 27]; *Figure 2L–N*). While the lack of additive effects is consistent with a linear InR/TOR pathway, we cannot completely rule out the possibility that an additional pathway downstream of InR might contribute to Dm8's dendritic development. Taken together, these findings suggest that the canonical InR/PI3K/TOR pathway positively regulates the dendritic field size of Dm8 neurons.

## SREBP is required for Dm8 dendritic elaboration

TOR forms two distinct multiprotein complexes referred to as TOR complex 1 (TORC1) and 2 (TORC2) to regulate a broad spectrum of cellular processes. TORC1 is composed of TOR, Raptor and LST8, whereas TORC2 contains TOR, Rictor, LST8 and Sin1 (*Sarbassov et al., 2005*; *Wullschleger et al., 2006*; *Bhaskar and Hay, 2007*). To dissect whether TORC1 or TORC2 is involved in Dm8 dendritic patterning, we generated *raptor*[del] and *rictor*[Δ2] null mutant Dm8 MARCM clones (*Li et al., 2019*; *Hietakangas and Cohen, 2007*). We found that *raptor*, but not *rictor*, mutant Dm8s exhibited significant reduction of dendritic field size (10.3 ± 1.5 dfu [n = 30] for *raptor*[del]; *Figure 2K,N*; 13.6 ± 0.3 dfu [n = 14] for *rictor*[Δ2], *Figure 2—figure supplement 1D*), suggesting that Dm8 dendritic expansion is regulated by TORC1 but not TORC2. To identify TORC1 targets involved in Dm8 dendritic expansion, we examined a number of known downstream molecules, including Atg7 (Autophagy-related 7), s6k (S6 kinase), Thor (eIF4E-binding proteins), Dref (DNA replication-related element factor) and SREBP (Sterol regulatory element binding protein) (*Porstmann et al., 2008*; *Parisi et al., 2011*). We found that *Srebp* mutants in particular had a reduced dendritic field size (*srebp*[189] 10.9 ± 0.3 dfu [n = 16], *Figure 2F and N*) compared to the wild-type. In contrast, mutations of the other genes did not show significantly reduced Dm8 dendritic field sizes (13.4 ± 0.2 dfu for *s6k*[l-1] [n = 16]; 13.3 ± 0.8 dfu for *Thor*[k07736] [n = 12]; 14.0 ± 0.6 dfu for *Atg7*[d06996] [n = 8]; 13.6 ± 0.6 dfu for *Dref*[KG09294] [n = 12]; *Figure 2—figure supplement 1E-J*). These findings indicate that the known TORC1 downstream target, SREBP, is cell-autonomously required for proper dendritic development of Dm8 neurons.

To further examine the relationship between TOR and SREBP in Dm8 dendrite development, we tested genetic epistasis and found that overexpressing the wild-type SREBP (SREBP[WT]) in *Tor* mutant Dm8 neurons restored normal dendritic development (12.8 ± 0.4 dfu [n = 14] for *Tor*[ΔP]/SREBP[WT]; *Figure 2I and N*). This result suggests that SREBP functions downstream of TOR to regulate Dm8 dendrite development (*Figure 2O*). In contrast, overexpressing a constitutively active version of SREBP (SREBP[CA]) in *Tor* mutant Dm8s resulted in very few recovered MARCM clones with *Tor*-like dendritic fields (10.0 ± 0.4 dfu [n = 4]; *Figure 2J and N*), suggesting cell toxicity of SREBP[CA]. While overexpressing SREBP[WT] in the wild-type background had little effect on Dm8 dendritic field sizes (13.1 ± 0.3 dfu [n = 14], *Figure 2G*), expression of SREBP[CA] led to decreased dendritic field sizes (9.8 ± 0.3 dfu [n = 8], *Figure 2H*), consistent with the results from a previous study that showed excessive SREBP activity is detrimental to dendrite development (*Ziegler et al., 2017*). SREBP is a transcription factor that enhances the expression of cholesterol and fatty acid biosynthesis enzymes, including acetyl-CoA carboxylase and fatty acid synthase (*Seegmiller et al., 2002*; *Liu et al., 2015*; *Camargo et al., 2009*; *Dobrosotskaya et al., 2002*; *Shao and Espenshade, 2012*). Together our data suggest that TORC1 regulates Dm8 dendrite development through SREBP-mediated transcriptional regulation of lipid synthesis (*Figure 2O*).

## Disrupted insulin/Tor signaling restricts dendritic fields and alters synapses

The altered dendritic field sizes in InR/Tor/SREBP mutants raised the question of whether the dendrites of mutant Dm8 neurons form functional synapses with appropriate numbers of R7 photoreceptors. The traditional GRASP (GFP reconstitution across synaptic partners) method reports membrane contacts between pre- and post-synaptic partners while the newer synaptobrevin-GRASP (syn-GRASP) method is capable of detecting active synapses (*Feinberg et al., 2008*; *Gordon and Scott,*

*2009*; *Macpherson et al., 2015*). To improve the specificity of synapse detection, we developed a 'receptor-based' version of the syn-GRASP method, called R-synGRASP, in which we replaced the membrane-tethered split-GFP component (CD4::GFP$^{sp11}$) with one tethering the split-GFP component (GFP$^{sp11}$) to neurotransmitter receptors, thereby increasing the specificity of the GRASP signals. Because Dm8 neurons receive histaminergic R7 inputs via the histamine-gated chloride channel, Ort (*Gao et al., 2008*), we engineered GFP$^{sp11}$::HA::Ort, in which the spGFP11 component and an HA tag were fused to immediate downstream of the Ort signal peptide. Two-electrode voltage clamp (TEVC) recordings on *Xenopus* oocytes expressing either wild-type Ort or GFP$^{sp11}$::HA::Ort confirmed that the fusion protein (GFP$^{sp11}$::HA::Ort) retained normal histamine-gated channel properties (*Figure 3—figure supplement 1A*). Furthermore, when expressed in Dm8 neurons, GFP$^{sp11}$::HA::Ort appeared to be enriched at the contacts between R7 and Dm8 dendrites with no detectable consequences on dendritic morphology or size (Supplementary *Figure 3—figure supplement 1D′*).

To demonstrate the efficiency of this receptor-based GRASP method (R-GRASP; *Figure 3A*), we first checked the membrane contacts between R7 and Dm8 neurons by expressing a membrane-tethered spGFP1-10 (CD4:: GFP$^{sp1-10}$) in all R7 neurons and the GFP$^{sp11}$::HA::Ort in single-cell Dm8 MARCM clones. In wild-type flies, multiple strong GRASP signals were detected at all apparent contact sites between Dm8 dendrites and R7 axonal terminals in the M6 layer (13.2 ± 0.4 columns, [n = 10] *Figure 3A, B and B′*), while no fluorescence was seen in flies that the expressed either of the split-GFP components alone (*Figure 3—figure supplement 1B and C* and data not shown). In contrast, *Tor* and *chico* mutant Dm8 neurons displayed GRASP signals at fewer R7 axon terminals (8.0 ± 0.4 columns for *Tor*, [n = 10]; 8.1 ± 0.3 columns for *chico* [n = 10]; *Figure 3C, C′, E and E′*), while *Pten* mutant Dm8 neurons showed GRASP signals at an increased number of R7 axon terminals (19.8 ± 0.4 columns, [n = 10] *Figure 3D and D′*) compared to wild type. The number of R7 contacts for wild-type and mutant Dm8 neurons correlated well with the size of dendritic fields, suggesting that the R-GRASP method faithfully reports membrane contacts.

To monitor *bona fide* active synapses using the R-synGRASP method, we expressed the Syb::GFP$^{sp1-10}$ in R7 neurons and the GFP$^{sp11}$::HA::Ort in single Dm8 MARCM neurons, and examined the native fluorescence signals of the reconstituted GFP (*Figure 3F*). In wild types exposed to a 12 hr light-dark cycle, the GRASP signals first emerged at the center of the Dm8 dendrites (corresponding to the R7 terminals in the home column), gradually becoming stronger and spreading to the periphery as the flies matured to 6 and 12 days of age (*Figure 3G, G′, L, L′ and K*). Meanwhile, wild-type files kept in constant darkness for 12 days showed essentially no GRASP signals (*Figure 3P and Q*). The light- and age-dependence of the GRASP signals suggests that the signals correspond to active synapses, consistent with the conclusions of the synaptobrevin-GRASP study (*Macpherson et al., 2015*). Notably, the most prominent GRASP signals were always present in the center of the Dm8 dendrites (*Figure 3G and L*), consistent with previous EM reconstruction studies that showed the center R7 (home) column is the major presynaptic partner for Dm8 neurons (*Gao et al., 2008*; *Takemura et al., 2013*). In 12-day-old wild-type flies reared under a 12 hr light-dark cycle, the GRASP signals corresponded to the same approximately 14 R7 terminals that were covered by the Dm8 dendrites (*Figure 3K and L*), suggesting that the peripheral synapses are also active. The slow emergence of the GRASP signals corresponding to peripheral synapses might have been due to the relatively low detection efficiency of the synaptobrevin-GRASP method.

Next, we examined active synapses in mutant Dm8 neurons devoid of various insulin/Tor signaling components. We found that the GRASP signals were completely absent in *Tor* mutant Dm8 neurons (*Figure 3H, H′, M, M′ and K*) and barely detectable in *chico* mutants in 3- to 12-day-old flies reared under a 12 hr light-dark cycle (*Figure 3J, J′, O and O′*). In contrast, in *Pten* mutant Dm8 neurons of 3- to 12-day-old flies, the GRASP signals were stronger in both the center and peripheral regions of the dendritic field, as compared to the wild types of the same age (*Figure 3I, I′, N and N′*). Notably, the GRASP signals for flies reared in constant darkness up to 12 days were even present in *Pten* mutant Dm8 neurons (*Figure 3R–T*), suggesting that the upregulation of TOR signaling elevates spontaneous synaptic activity. To check for potential structural defects, we examined the localization of pre- and post-synaptic markers in wild-type and mutant flies. We found that compared to wild types, *Tor* mutant (but not *chico* or *Pten* mutant) Dm8 neurons had reduced Ort expression (*Figure 3—figure supplement 1D′-E′*), suggesting that Tor is required for the production or membrane trafficking of Ort. On the other hand, we found no detectable changes in the presynaptic protein markers, Syb and Brp (Bruchpilot), in the R7 terminals that made apparent contacts with *Tor* and

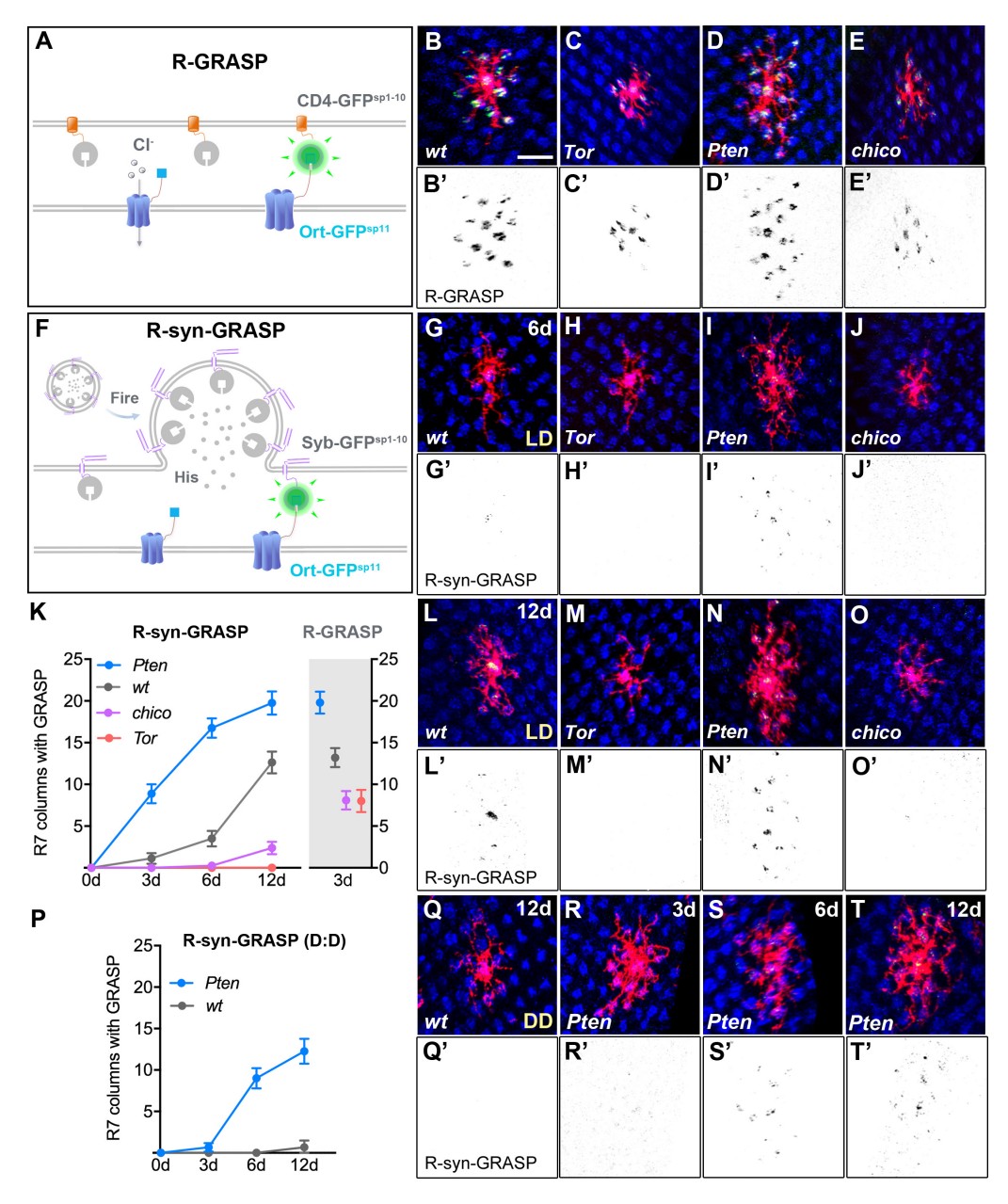

**Figure 3.** Defects in Dm8 dendritic fields caused aberrant synaptic connections with R7 photoreceptors. (**A–E**) Defective dendrites of *Tor*, *Pten* and *chico* mutant Dm8 neurons exhibit altered membrane contacts with R7 photoreceptors, revealed by the 'receptor-based' version of GFP Reconstitution Across Synaptic Partners technique (R-GRASP). (**A**) A schematic illustration of R-GRASP is shown. Membrane tethered spGFP1-10 (CD4-GFP^sp1-10) (grey) is expressed in presynaptic neurons (R7s), while the other GFP fragment (blue), spGFP11 fused to histamine receptor Ort (Ort-GFP^sp11), is expressed at postsynaptic neurons (Dm8). Functional reconstitution of the two GFP fragments generates fluorescent GFP (green), indicating neuronal contact sites. (**B–E**) The R-GRASP technique was applied to monitor neuronal contacts between R7 photoreceptors and their postsynaptic Dm8 partners in single wild-type (**B, B'**), *Tor* mutant (**C, C'**), *Pten* mutant (**D, D'**) and *chico* mutant (**E, E'**) Dm8 neurons. The photoreceptor R7 axons were labeled with anti-24B10 (blue), and the Dm8 MARCM clones were labeled by mCD8Cherry (red). The R-GRASP signal between R7 and single Dm8 cells is shown in green. *Tor* (**C, C'**) and *chico* (**E, E'**) mutations in Dm8 neurons diminished the number and area of contact with R7 cells, while *Pten* (**D, D'**) mutation in Dm8 neurons significantly increased the contact number. (**F**) A schematic illustration of the neurotransmitter 'receptor-based' version of synaptobrevin GRASP (R-synGRASP). Upon neuronal firing, activity-dependent synaptobrevin-fused spGFP1-10 (Syb-GFP^sp1-10) in synaptic vesicles become localized to the

*Figure 3 continued on next page*

*Figure 3 continued*

presynaptic sites and exposed to the Ort-GFP$^{sp11}$ expressed at postsynaptic sites. Reconstitution of the two GFP fragments in the synaptic cleft generates functional GFP to indicate active synapses. (G–O) Dendritic defects in *Tor*, *Pten* and *chico* mutant Dm8 neurons caused synaptic defects, as revealed by R-synGRASP under a 12 hr light-dark cycle (L:D). R-synGRASP between R7 and Dm8 neurons was monitored in 6-day-old (G, G') and 12-day-old (L, L') wild-type flies, 6-day-old (H, H') and 12-day-old (M, M') *Tor* mutants, 6-day-old (I, I') and 12-day-old (N, N') *Pten* mutants, and 6-day-old (J, J') and 12-day-old (O, O') *chico* mutant Dm8 neurons. (G, G', L, L') 6-day-old and 12-day-old wild-type flies exhibited the most prominent GRASP signal in the central R7 column. (H, H', M, M') *Tor* mutation in Dm8 neurons caused a complete loss of active synapses with R7 photoreceptors. (I, I', N, N') *Pten* mutation in Dm8 neurons resulted in aberrant synapse formation with R7 photoreceptors. (J, J', O, O') *chico* mutant Dm8 neurons exhibited significantly decreased number of active synapses with R7 photoreceptors. (K) Quantification of R-synGRASP in 0- to 12-day-old flies and quantification of R-GRASP in 3-day-old flies under a 12 hr light-dark cycle (L:D cycle). *Pten* mutant Dm8 neurons displayed R-synGRASP signal in consistently more R7 columns, while *chico* and *Tor* mutants exhibited R-synGRASP signal in fewer R7 columns. The R-synGRASP signals in *Pten* mutant and wild-type 12-day-old flies were comparable to R-GRASP in 3-day-old flies. (P–T) *Pten* mutant Dm8 neurons form synapses with R7 photoreceptors under constant darkness (D:D cycle). (P) R-synGRASP signal was quantified in wild-type and *Pten* mutant Dm8 neurons of flies raised in constant darkness. R-synGRASP signal was nearly absent in 12-day-old wild-type neurons (Q, Q'). In contrast, GRASP signal appeared in 3-day-old *Pten* mutant Dm8 neurons (R, R') and became stronger in 6-day-old (S, S') and 12-day-old (T, T') adult flies. (B'–E', G'–J', L'–O', Q'–T') GRASP signals shown in inverted black-and-white images for (B–E, G–J, L–O, Q–T), respectively. The online version of this article includes the following figure supplement(s) for figure 3:

**Figure supplement 1.** In vitro TEVC recording of GFP$^{SP11}$-2xHA-Ort and its in vivo expression in wild-type and mutant Dm8 clones in adults.

*Pten* mutant Dm8 neurons (*Figure 3—figure supplement 1I''',1J'''*). As R7s are presynaptic chiefly to Dm8s but also to other medulla neurons, we could only conclude that R7 presynaptic sites did not drastically change. Together, these findings suggest that defective insulin/Tor signaling alters synaptogenesis or synaptic function of Dm8 neurons in addition to dendritic field size.

## Endogenous insulin receptors are localized to Dm8 dendrites

The observation that insulin signaling regulates Dm8 dendritic development and synaptic activity raised the question of where within the neuron do these signaling events take place. To answer this question, we examined the subcellular localization of InR in developing Dm8 neurons by tagging endogenous InR with a V5 epitope and three copies of split-GFP11 (GFP$^{SP11}$; *Cabantous et al., 2005*; *Kamiyama et al., 2016*) using CRISPR/Cas9 (*InR::V5::GFP $^{SP11}$*, *Figure 4A*). The resulting *InR:: V5::GFP $^{SP11}$* allele is a fully viable and functional *InR* allele. While the presence of the V5 epitope allowed us to examine endogenous InR patterns in whole animals, we further generated an UAS-FRT-Stop-FRT-GFP$^{SP1-10}$-T2A-myr::tdTomato line, which enables conditional expression of the cyto-solic split-GFP1-10 (GFP$^{SP1-10}$) and the membrane marker myr::tdTomato in single mosaic cells (*Figure 4B*). With heat-shock-induced expression of flippase recombinase and appropriate Gal4 drivers, this split-GFP tagging strategy allowed us to visualize the reconstituted GFP (endogenous InR) in single neurons marked with membrane-tethered tdTomato (*Figure 4C*).

We first used anti-V5 staining to examine the overall InR pattern in the developing brains at the pupal stage, approximately 20 hr after puparium formation (APF). We found that InR is enriched in the neuropil where axonal terminals and dendrites are located, in addition to neuron cell membranes in the medulla cortex (*Figure 4D and D'*). We further examined the subcellular localization of InR in photoreceptors using the photoreceptor-specific flipase transgene and the split-GFP tagging method. The reconstituted GFP signals indicated that InRs were enriched in the growth cones of R7 and R8 photoreceptors in 30 and 60 hr APF brains (*Figure 4—figure supplement 1A-C'*). This result was consistent with a previous study that suggested InR is localized to photoreceptor axonal terminals, where it serves as a guidance receptor (*Song et al., 2003*).

Next, we examined the location of InR in single developing Dm8 neurons by flip-out induced expression of GFP$^{SP1-10}$ and the membrane marker myr::tdTomato at 40–70 hr APF, the timing at which Dm8 neurons elaborate their dendritic arbors (*Ting et al., 2014*) and in adults. We observed strong GFP fluorescence puncta in Dm8 dendritic arbors at 40 hr APF in flies carrying the *InR::V5:: GFP $^{SP11}$* locus (*Figure 4E, E', F and F'*). In contrast, very weak fluorescence signals were observed

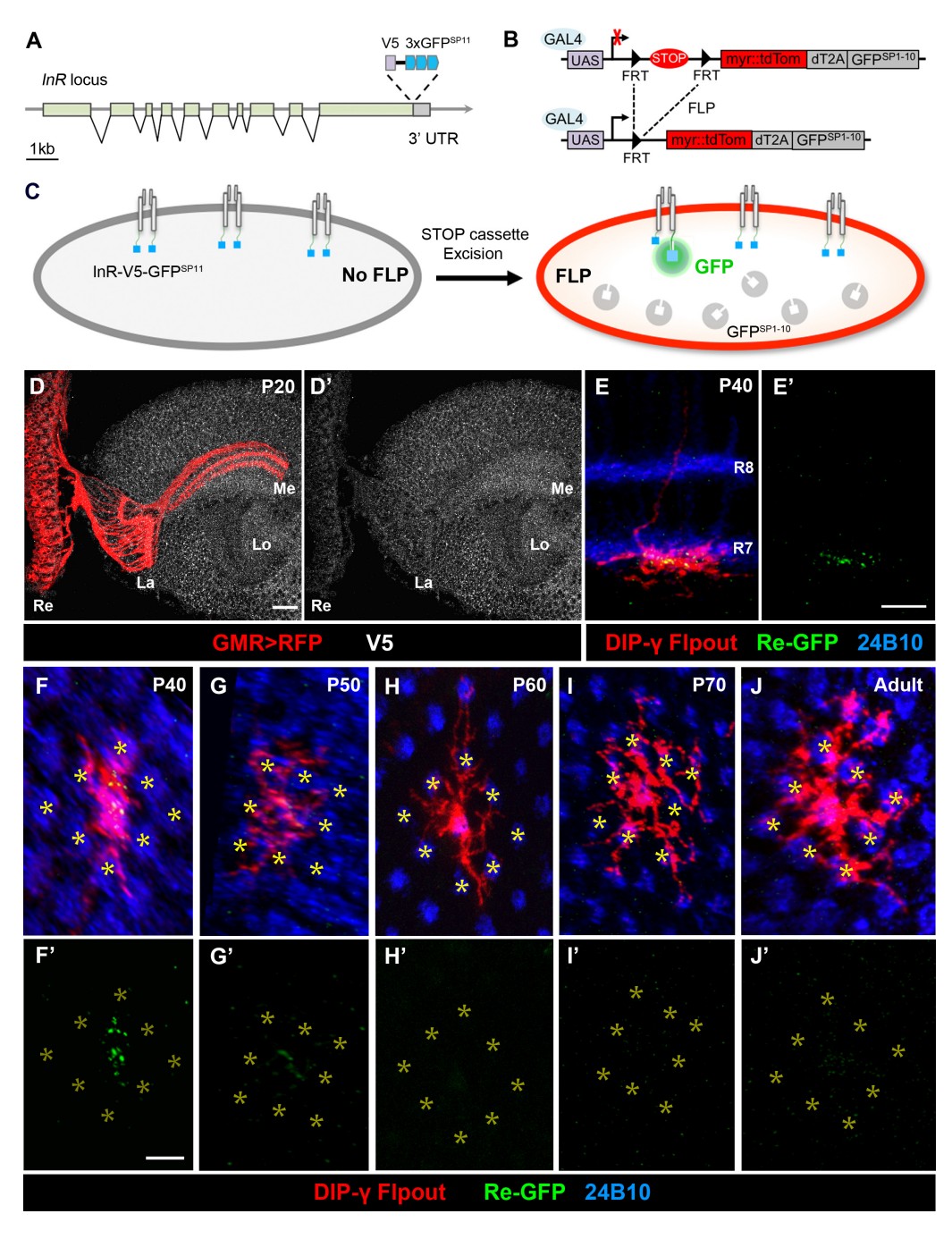

**Figure 4.** Split-GFP tagging of insulin receptor reveals its endogenous expression pattern in the optic lobe and Dm8 dendrites. (A–C) Schematics depict the split-GFP tagging strategy to visualize endogenous insulin receptor (InR) expression at single cell resolution. (A) Three copies of split-GFP11 (3x GFP$^{SP11}$, blue) and a V5 epitope (grey) were inserted into the end of the coding region of the *InR* locus, immediately before the stop codon. (B) Heat-shock-induced expression of flippase recombinase (FLP) excised the STOP cassette flanked by two FRTs, allowing the targeted neurons to express membrane-tethered tdTomato (myr::tdTom) and split-GFP1-10 (GFP$^{SP1-10}$) in the cytosol. (C) After excision of the STOP cassette, the targeted neurons express tdTom and GFP$^{SP1-10}$. The cytosolic GFP$^{SP1-10}$ binds to GFP$^{SP11}$ on the intracellular C-terminus of InR, constituting a functional GFP to indicate the location of the endogenous InR. (D–D') Endogenous InR in the optic lobe was visualized by anti-V5 staining (white) in developing pupae at 20 hr APF (after puparium formation). The photoreceptors were labeled by GMR-Gal4-driven RFP (red). InR was highly expressed in the cell bodies of visual cortex neurons and the neuropils of neurons

*Figure 4 continued on next page*

*Figure 4 continued*

in the visual system. Re, retina; La, lamina; Me, medulla; Lo, lobula. Scale bar, 10 μm. (**E–E'**) Endogenous InR is highly expressed in the central area of Dm8 dendrites at 40 hr APF. tdTom (red) labels a single Dm8 flipout clone, which simultaneously expresses GFP^sp1-10 after STOP cassette excision. The photoreceptors R7/8 layers and growth cones of R7 were labeled by anti-24B10 (blue). InR, indicated by the prominent GRASP signals (green), was located in the central region of Dm8 dendrites, shown in dorsoventral views. Scale bar, 5 μm. (**F–J'**) The temporal patterns of endogenous InR expression in Dm8 dendrites surrounded by eight adjacent R7 growth cones (yellow asterisks), shown in proximodistal views. Single Dm8 neurons were labeled by DIP-γ-Gal4 with Flp-FRT flipout system (red). The R7 photoreceptors were labeled with anti-24B10 (blue) and GRASP signals (green). InR is strongly expressed in the central area of Dm8 dendrites at 40 hr APF (**F, F'**), subsides at 50 hr APF (**G, G'**) and is absence at 60 hr (**H, H'**), 70 hr (**I, I'**) APF and adult (**J, J'**).

The online version of this article includes the following figure supplement(s) for figure 4:

**Figure supplement 1.** InR is expressed in the growth cones of photoreceptors during pupal stages.

---

in Dm8 dendrites with GFP^SP1-10 expression but without the knock-in InR locus (data not shown), consistent with a low level of background GFP^SP1-10 fluorescence (*Leonetti et al., 2016*; *Feng et al., 2017*). Strikingly, at 40 hr APF, the reconstituted GFP signals indicating endogenous InR were concentrated in the central area of Dm8 dendrites, corresponding to the home column, without observable signal at the peripheral dendrites (*Figure 4F'*). The reconstituted GFP signals in Dm8 dendrites subsided at 50 hr APF and were absent beyond 60 hr APF and in adults (*Figure 4H–J'*). Thus, our findings suggested that endogenous InR is concentrated in the central region of Dm8 dendrites at 40 hr APF and appears to be absent in Dm8 dendrites at and after 60 hr APF and in adults.

## Lamina L5 neurons provide DILP2 to Dm8 dendrites

The presence of InR on developing Dm8 dendrites suggests that it might be activated by local ligands. A recent RNA-seq study revealed high transcript levels for the insulin-like peptide DILP2 in developing L3 and L5 lamina neurons (*Tan et al., 2015*). We also noted that from 20 hr APF to adult stages, L5 growth cones are adjacent to R7 photoreceptor growth cones and are therefore in close proximity to developing Dm8 dendrites (*Figure 5A and G,G'*; *Nern et al., 2008*; *Ting et al., 2014*). We further confirmed *dilp2* expression in L5 neurons at 40 hr APF by in situ hybridization (*Figure 5D–E'*). To test whether lamina neurons (LNs) can transport DILP2 to their growth cones, we expressed a GFP-tagged DILP2 in the LNs using the pan-LN driver, GM9B08-Gal4 (*Wong et al., 2012*; *Pecot et al., 2013*). Strong GFP signals were present in developing LN growth cones (*Figure 5B–C'*), suggesting that developing LNs, like motor neurons, are capable of transporting DILP2 to their axonal terminals (*Wong et al., 2012*).

We next examined whether DILP2 is functionally required in LNs for Dm8 dendritic development using transgenic RNAi. We knocked down *dilp2* in developing LNs using a previously characterized UAS-*dilp2* RNAi transgene and two different pan-LN drivers, GMR9B08-Gal4 and GMR27G05-Gal4, after which we examined Dm8 dendritic arborization in adults. To enhance RNAi potency, we included a UAS-Dicer-2 (Dcr2) transgene and raised the animals at 28°C. We found that Dm8 dendritic fields were significantly smaller in LN *dilp2* RNAi knock-down animals (10.1 ± 0.5 dfu [n = 16] for *9B08 > dilp2* RNAi; 12.1 ± 0.4 dfu [n = 15] for *27G05 > dilp2* RNAi; *Figure 5I, J and P*) compared to the control, but the layer-specific targeting of Dm8 dendrites was unaffected. In addition, we observed disrupted R7 axonal tiling in the *9B08 > dilp2* RNAi animals (*Figure 5I*; *Figure 5—figure supplement 1C–C''*). In contrast, knockdown of *dilp2* or *dilp6* in all photoreceptors using the pan-photoreceptor GMR-Gal4 driver did not cause any alteration in Dm8 dendritic patterns or R7 tiling (13.4 ± 0.5 dfu [n = 14] for *dilp2-RNAi*; 13.3 ± 0.4 dfu [n = 9] for *dilp6-RNAi*; *Figure 5L, M and P*; *Figure 5—figure supplement 1C–D*). Together, these data indicate that knockdown of *dilp2* in LNs, but not in photoreceptors, decreases the size of Dm8 dendritic fields and causes R7 axonal tiling defects.

We next selectively knocked down *dilp2* in L5 neurons using the L5-specific driver 6–60 Gal4 and examined Dm8 dendritic patterning in adult animals. We found that *dilp2* knockdown in L5 caused a significant reduction of Dm8 dendritic field size (11.1 ± 0.4 dfu [n = 20]; *Figure 5K and P*), without affecting layer-specific targeting of Dm8 dendrites or R7 axonal tiling (*Figure 5—figure supplement 1B–B'''*). Conversely, overexpressing DILP2 in L5 neurons only marginally increased the size of Dm8

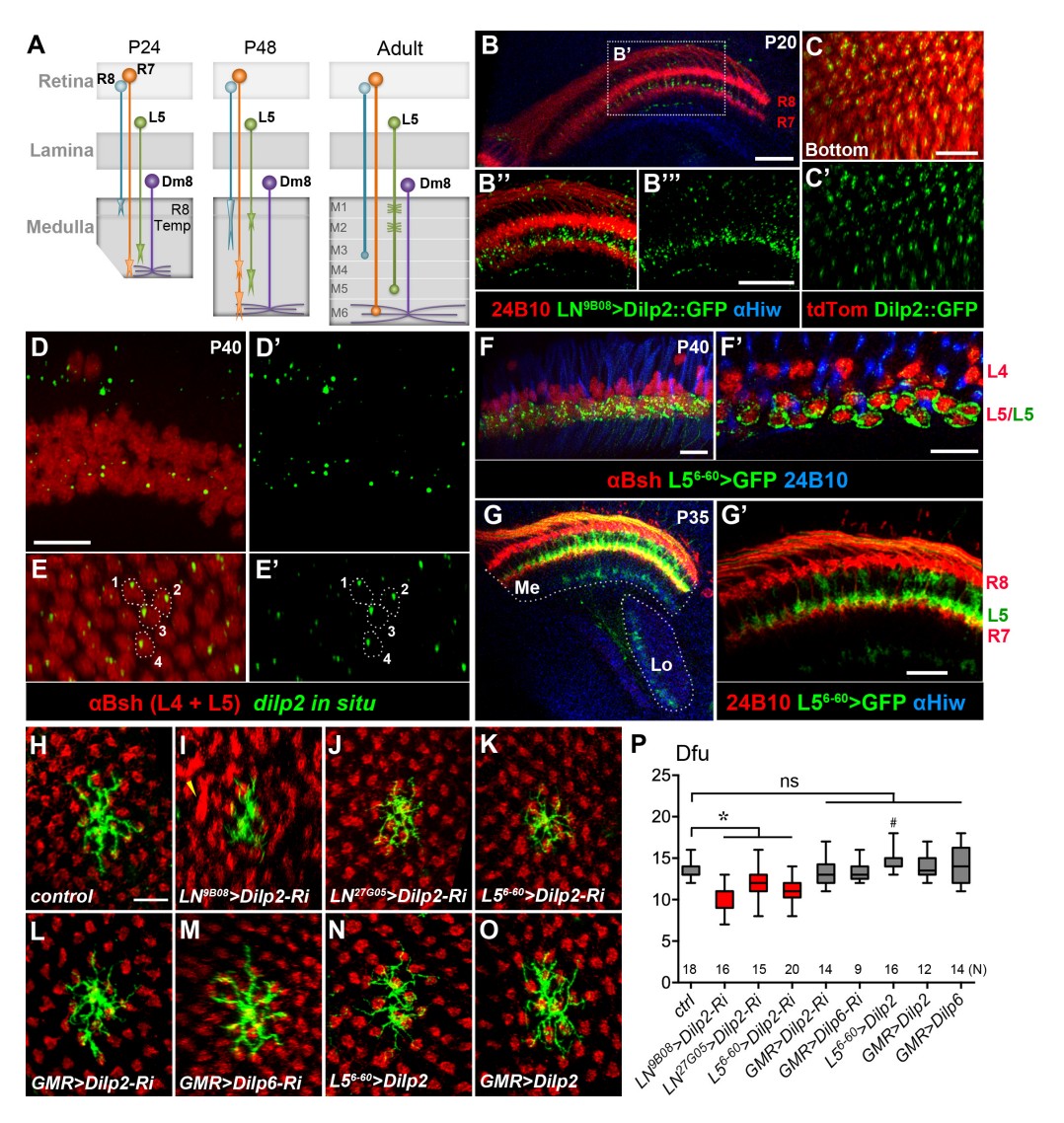

**Figure 5.** DILP2 derived from L5 lamina neurons positively regulates Dm8 dendritic field elaboration. (**A**) A schematic illustration showing the developmental processes of R7 and R8 photoreceptors, L5 lamina neurons and Dm8 neurons. A Dm8 neuron (purple) expands its dendritic arbors during development to reach its final size. R7 (orange), R8 (cyan) and L5 (green) growth cones undergo dynamic changes at their target locations. R8 growth cones first project to a temporary layer at 24 hr APF and then extend to a deeper layer (M3 in the adult). At 24 hr APF (after puparium formation), R7 growth cones first target the deep medulla layer where Dm8 dendrites reside, and then reshaped and stabilized at the M6 layer in the adult. L5 growth cones extend close to Dm8 dendrites by 24 APF and eventually to the M5 layer at the adult stage. (**B–B"'**) Overexpression of a GFP-tagged DILP2 (green) in LNs led to its accumulation in the deep layer of lamina neurons. Photoreceptors were labeled by anti-24B10 (red), and medulla (Me) and the lobula (Lo) neuropil were labeled by anti-Highwire (Hiw) (blue). Scale bar, 20 μm. (**C–C'**) The GFP-tagged DILP2 (green) is present in the growth cone of lamina neurons (red). The bottom (proximodistal) view is shown. Scale bar, 10 μm. (**D–E'**) In situ hybridization of *dilp2* in the developing pupal eye disc. In situ hybridization detected a high level of *dilp2* mRNA (green) in the developing L5 lamina neurons (red) which were labeled by anti-Bsh immunostaining (red). Anti-Bsh also labeled the nuclei of L4 lamina neurons (the upper row) in addition to L5's (the lower row). Side view (**D, D'**) and bottom view (**E, E'**) are shown. The *dilp2* mRNA was located in the cytosol adjacent to the nucleus of L5, as shown in E and E'. Scale bar, 10 μm (in D). (**F–F'**) The L5 cell bodies were labeled using 6–60 Gal4 driving the mCD8GFP marker (green). Photoreceptors were labeled by anti-24B10 (blue) as landmarks. (**F'**) shows a single z-section in (**F**). Scale bar, 10 μm. (**G–G'**) 6–60 Gal4 labeled L5 processes (green) in the medulla (Me), anti-24B10 staining R7/R8 (red) and anti-Hiw (blue) for neuropil landmarks. Scale bar, 10 μm (in G'). 6–60 Gal also labeled few trans-medulla neurons with processes in the medulla and lobula. (**H–O**)

*Figure 5 continued on next page*

*Figure 5 continued*

RNAi-mediated *dilp2* knockdown in pan-lamina neurons or L5s caused Dm8 dendritic field reduction, as examined in adults. Single flipped-out Dm8 clones (green) with *dilp2*-RNAi alone (H) or driven two pan-LN driver, 9B08-Gal4 (I) and 27G05-Gal4 (J), or L5-specific 6–60 Gal4 (K), or photoreceptor-specific GMR-Gal4 (L). Misalignment of photoreceptor terminals was consistently observed with pan-LN knock-down of *dilp2* (yellow arrow). (M) GMR-Gal4-driven *dilp6-RNAi* had minimal effects on Dm8 dendrites. (N–O) Overexpression of *dilp2* in L5s or photoreceptors did not significantly alter Dm8 dendritic field sizes. Scale bar, 10 μm (in H). (P) Box plot showing the dendritic field sizes of Dm8 neurons in wild-type and *dilp* manipulations. *Dilp2* knock-down in pan-lamina neurons or L5 neurons specifically reduced the Dm8 dendritic fields. *p<0.05; #p=0.002; ns, not significant, unpaired Student's t-test. N; the number of cells scored for each genotype.

The online version of this article includes the following figure supplement(s) for figure 5:

**Figure supplement 1.** RNAi-mediated knockdown of *dilp2* in lamina neurons caused tiling phenotype of photoreceptor R7.

---

dendritic fields (14.6 ± 0.4 dfu [n = 16]; *Figure 5N and P*), while overexpressing DILP2 or DILP6 in photoreceptors did not affect Dm8 patterning (*GMR > Dilp2*, 13.9 ± 0.5 dfu [n = 12]; *GMR > Dilp6*, 14.3 ± 0.6 dfu [n = 14]; *Figure 5O and P*). In summary, *dilp2* in L5 neurons is specifically required for appropriate sizing of Dm8 dendritic fields.

## Insulin and activin act in parallel pathways to control Dm8 dendrite development

We have previously shown that R7-derived Activin signals through the receptor, Baboon, to restrict Dm8 dendritic fields, and removing R7 photoreceptors (the source of Activin) or Baboon in Dm8 neurons resulted in aberrant expansion of dendritic fields (*Ting et al., 2014*). Thus, Dm8 neurons appear to receive both positive (insulin) and negative (Activin) signals to regulate dendritic development. To determine how these two seemingly opposing signals converge in Dm8 neurons, we carried out a genetic interaction test by disrupting both signaling pathways. We found that disrupting both Activin and insulin signaling resulted in additive effects on Dm8 dendritic development; in the absence of R7 photoreceptors (hence the source of Activin), *Tor* mutant Dm8 neurons have an average dendritic field size (13.1 ± 0.4 dfu [n = 25]; *Figure 6B and K*) comparable to wild type. Similarly, ectopically expressing a dominant-negative form of Baboon (Babo^DN) in single *Tor* mutant Dm8 neurons resulted in an average dendritic field size (13.5 ± 0.4 dfu [n = 28], *Figure 6D and K*) close to wild type. Interestingly, even though the average size of the dendritic fields were similar to wild type, mutant Dm8 neurons lacking both Activin and insulin signaling appeared to have more variation in dendritic field sizes (*Figure 6B, D and E*), an observation that was confirmed by statistical testing (*sev/Tor vs. wt*: p=0.0004 for Conover Test; p=0.0045 for Siegel-Tukey Test; Babo^DN/*Tor vs. wt*: p=0.0034 for Conover Test; p=0.0343 for Siegel-Tukey test). Together, these findings suggest that insulin/Tor and Activin/Babo signal in parallel pathways in Dm8 neurons to ensure the development of appropriate and consistently sized dendritic fields.

We next determined how simultaneously activating TOR signaling and inhibiting Activin/Babo signaling pathways would affect Dm8 dendritic fields. We found that removing R7 neurons (the source of Activin) using the *sevenless* mutation modestly increased the dendritic field sizes of *Pten* mutant Dm8 neurons (21.6 ± 0.5 dfu [n = 8] for *sev/Pten*) while expressing Babo^DN had no significant effect (20.9 ± 0.3 dfu [n = 34] for Babo^DN/*Pten*; *Figure 6G, H and K*). Similarly, removing R7 neurons did not significantly affect the dendritic sizes of *Tsc1* mutant Dm8 neurons (21.3 ± 0.7 dfu [n = 10]; *Figure 6J and K*). This weak or lack of additive effect could have been due to an absence of genetic interaction or ceiling effects. To differentiate between these two possibilities, we tested whether over-activating Activin/Baboon signaling could affect dendritic development of *Pten* mutant Dm8 neurons. We found that overexpressing a dominant active form of Babo (Babo^DA) in *Pten* mutant Dm8 neurons resulted in smaller dendritic field sizes (19.3 ± 0.4 dfu [n = 12]) than those in *Pten* mutants alone (*Figure 6I and K*). These results are consistent with the notion that *Pten* and *babo^DA* have additive effects on Dm8 dendritic development, while simultaneous activation of TOR signaling and inhibition of Activin/Babo signaling causes Dm8 neurons to reach a near-maximal dendritic field size.

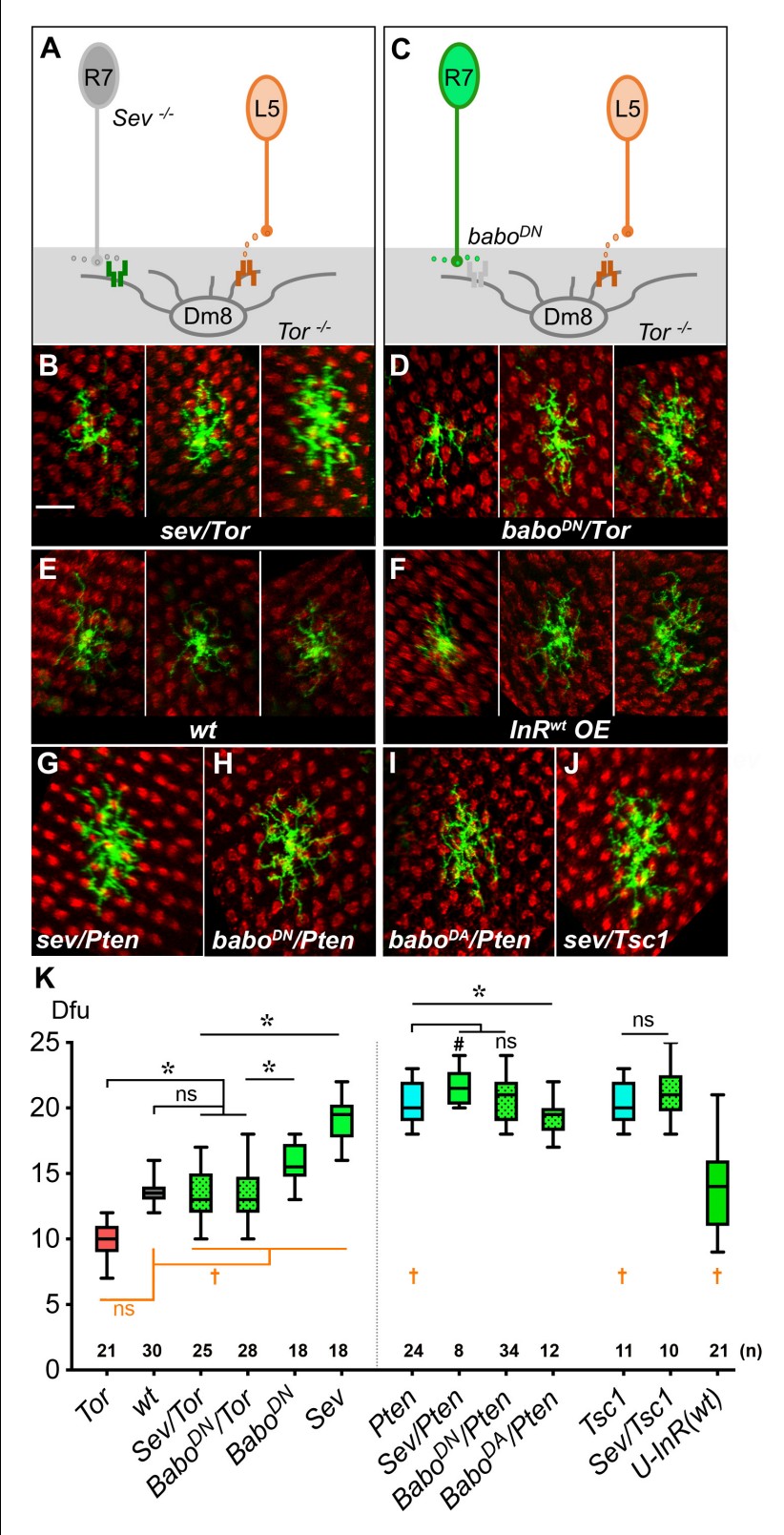

**Figure 6.** Insulin/Tor signaling operates in parallel with Activin/Baboon signaling to control Dm8 dendritic field sizes. (A–B) Blocking both insulin and Activin signaling in Dm8s resulted in an additive effect and highly variable dendritic field sizes. (A) An illustration depicts *Tor* mutant Dm8s, which receive but can't respond to L5-derived DILP2, in the *sevenless* (*sev*) mutant background which ablates R7s, hence the source of Activin. (B) Three

*Figure 6 continued on next page*

*Figure 6 continued*

examples of *Tor* mutant Dm8s in the *sev* background. The dendritic fields of *sev/Tor* mutants displayed great variation, spanning from 10 to 17 dfu, while the average was comparable to wild type. Single *Tor* mutant Dm8 neurons were generated using the Flp-MARCM system and labeled with mCD8GFP (green), and examined in adults. (C–D) Ectopic expression of a dominant negative *Babo* (*Babo^DN*) in *Tor* mutant Dm8 neurons recapitulated the variability phenotypes of the *sev/Tor* double mutation. (C) An illustration depicts that ectopic expression of *Babo^DN* in *Tor* mutant Dm8s (*Babo^DN/Tor*) prevents Dm8s from responding to both L5-derived DILP2 and R7-derived Activin. (D) Three examples of the *Babo^DN/Tor* mutant Dm8 neurons. *Tor* mutant Dm8s expressing the dominant negative form of Baboon (*Babo^DN*) were generated using the Flp-MARCM system and the DIP-γ-Gal4 driver. Dm8 dendritic fields of *Babo^DN/Tor* mutants are highly variable in sizes, spanning 10–18 dfu while the average size was comparable to wild type (E). (F) Expressing insulin receptors in single Dm8s throughout pupal stages led to highly variable dendritic field sizes. (G–J) Single Dm8 mutant clones were labeled with Flp-MARCM with various genetic manipulations on insulin and Activin signaling. (G) *Pten* mutant Dm8 in the *sev* background (*Sev/Pten*); (H) *Pten* mutant Dm8 ectopically expressing *Babo^DN*, a dominant negative form of Baboon (*Babo^DN/Pten*); (I) *Pten* mutant Dm8 expressing *Babo^DA*, a dominant active form of Baboon (*Babo^DA/Pten*); and (J) *Tsc1* mutant Dm8 in the *sev* background (*Sev/Tsc1*). The double mutants of *sev/Pten*, *Babo^DN/Pten*, and *Sev/Tsc1* resembled single mutations of *Pten* or *Tsc1*, while *Babo^DA/Pten* moderately reduced the *Dm8* dendritic field sizes as compared with *Pten* mutation. Photoreceptor axon terminals were labeled by anti-24B10 staining for landmarks (red). Scale bar, 10 µm (in B for B-J). (K) Box plot showing the dendritic field sizes of Dm8 neurons in wild type and mutants. Analysis of means (colored in black): *$p < 0.05$; ns, not significant, unpaired Student's t-test. Analysis of variance (in orange): †$p < 0.05$; ns, not significant, Conover test and Siegel-Turkey test.

---

Notably, while disrupting both Activin and insulin signaling caused highly variable dendritic field sizes with a near-wild-type average, removing negative regulators of the insulin signaling pathway (TSC1 and Pten) to disrupt its spatiotemporal restriction led to large and variable Dm8 dendritic fields (*Pten vs. wt*: $p = 0.0062$ for Conover Test; $p = 0.049$ for Siegel-Tukey Test; *TSC1 vs. wt*: $p = 0.0044$ for Conover Test; $p = 0.016$ for Siegel-Tukey test). Furthermore, expressing insulin receptors beyond its normal temporal window led to highly variable Dm8 dendritic fields (*U-InR vs. wt*: $p = 2.02E-6$ for Conover Test; *Figure 6F and K*), highlighting the importance of spatiotemporal restriction of InR expression.

## Stage-dependent alteration of dendritic development parameters results in large and consistently sized dendritic arborizations

Two parameters describing dendritic development, branching and terminating rates ($k_b$ and $k_t$), can largely determine the sizes and complexity of resultant dendritic arborizations. However, the specific effects of these parameters on robustness of dendritic field sizes are entirely unknown. Activin signaling increases the terminating rate and negatively affects dendritic arborization sizes (*Ting et al., 2014*); meanwhile, a positive effector, such as insulin signaling, might increase dendritic arborization sizes by reducing the terminating rate and/or increasing the branching rate. Because Dm8 dendritic arbors are too dense to trace, we could not experimentally determine the dendritic branching and terminating rates for Dm8s, as was previously done for Tm20s (*Ting et al., 2014*). Therefore, to gain insight into how Activin and insulin signaling affect the sizes and variability of Dm8 dendritic arborizations, we simulated dendritic growth with broad $k_b$ and $k_t$ parameters using a Monte Carlo approach (see Materials and methods for details). The radii of dendritic fields of simulated neurons were calculated as the distance of 95th percentile of all dendritic terminal points to the origin of growth (R95). For columnar Tm neurons, such as Tm2, Tm9 and Tm20, the branching and terminating rates are constant (*Ting et al., 2014*), implying linear kinetics in the increase and decrease of dendrite numbers. Using the branching and terminating rates reported for these neurons (0.360 and 0.594 µm$^{-1}$ for Tm20s; *Ting et al., 2014*), we found that the dendrites terminate 3.17 ± 0.69 µm (21.8% coefficient of variation; CV) from the origin, a distance that spans approximately one medulla column (*Figure 7D*). Using the parameters of *babo* mutant Tm20s (0.322 and 0.431 µm$^{-1}$ for $k_b$ and $k_t$; *Ting et al., 2014*), which have larger than wild-type dendritic fields, the simulated dendritic fields were 4.32 ± 1.11 µm (25.7% CV). In contrast, the large dendritic arborizations of Dm8 neurons normally contact about 14 R7 terminals and span about 14 columns, corresponding to a field of approximately 10 ~ 11 µm in diameter.

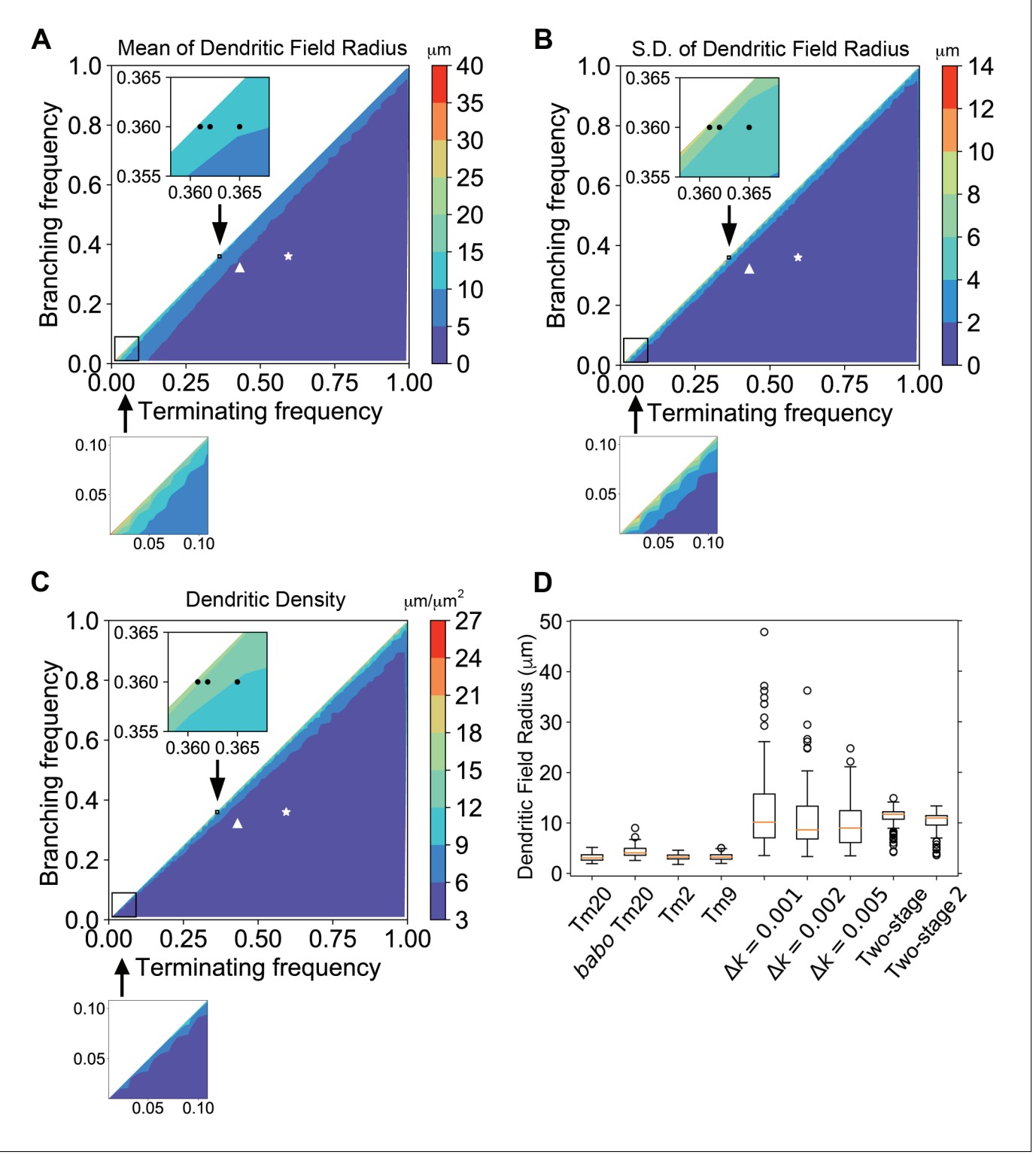

**Figure 7.** Dendritic growth simulation reveals the relationship between kinetic parameters and field sizes and variability. Stochastic dendritic growth was simulated with broad branching ($k_b$) and terminating ($k_t$) rate parameters and the radii of the resulting dendritic fields (R95) were calculated as the distance of 95th percentile of all terminal points to the origin. (A,B,C) Heat maps show the mean (A), standard variation (B) and density (C) in the R95 radius. Color bars display the color range as indicated. White stars and triangles indicate the position of reported parameters for wild-type and the *babo* mutant Tm20, respectively (*Ting et al., 2014*). Black dots in the zoomed inset indicate the trial parameter sets for generating large neurons, correspond to branching rate $k_b$ = 0.360 $\mu m^{-1}$ and terminating rate $k_t$ = 0.361, 0.362, and 0.365 $\mu m^{-1}$ (from left to right), respectively. In the bottom

*Figure 7 continued on next page*

*Figure 7 continued*
row are zoom-in for the lower-left corner for very low branching and terminating rates, with the same color cues as the corresponding top panels. (D) R95 distributions of 100 simulated neurons for Tm20 (wild type and *babo* mutant), Tm2, and Tm9 with their parameters reported previously, followed by tests with $k_b = 0.36\ \mu m^{-1}$ and $k_f = 0.361$, 0.362, and $0.365\ \mu m^{-1}$, respectively. For the two-stage model, $k_t$ was kept at $0.4\ \mu m^{-1}$ and $k_b$ started with 0.42 $\mu m^{-1}$ initially, and switched to $0.36\ \mu m^{-1}$ at the second stage; for the two-stage-2, the $k_t$ raises to $0.42\ \mu m^{-1}$ at the second stage.

The online version of this article includes the following figure supplement(s) for figure 7:

**Figure supplement 1.** Dendritic growth simulation with different numbers of initial dendrites.

We simulated dendritic growth for a broad parameter space to explore sets that could generate large dendritic field sizes (*Figure 7A–C*). We found that to approach the large field sizes observed for Dm8 dendrites with constant branching and terminating rates, $k_b$ and $k_t$ needed to be very close to each other, but this similarity also led to highly variable sizes (*Figure 7A,B*), presumably due to dynamic instability (*Mitchison and Kirschner, 1984*; *Goldbeter and Koshland, 1981*). When the Tm20 branching rate was increased by 10% and Tm20 terminating rate was reduced by 10%, the dendrite terminating distance became $5.89 \pm 3.17\ \mu m$; if the increase and decrease were 20%, the terminating distances became $13.83 \pm 10.59\ \mu m$. Thus, larger changes in the rates could increase the dendrite growth region, but this effect was accompanied by a major increase in variability. The CV (standard deviation/mean) was 21.8% for the original parameters, 54% for 10% changes, and 77% for 20% changes. We further examined the relationship between the $k_t$ - $k_b$ difference ($\Delta k$) and variability. With $k_t$ constant at $0.361\ \mu m^{-1}$ and $\Delta k = 0.005$, 0.002 and 0.001, the field diameters were $9.98 \pm 4.68$ (10.16 median, 46.9% CV), $10.79 \pm 6.09$ (8.64 median, 56.4% CV), and $12.74 \pm 8.41\ \mu m$ (9.00 median, 66.0% CV), respectively (*Figure 7D*). Thus, when the branching frequency approaches the terminating frequency, neurons can grow larger dendritic field sizes at a steep cost of variability (*Figure 7A and B*). Interestingly, when terminating and branching frequencies are very low, sizable dendritic fields may be generated with modest variability (*Figure 7A and B*). However, the dendritic arbors become sparse as the dendritic segments grew much longer before branching or terminating (average $24.3\ \mu m$ for $k_b = 0.02$ and $k_t = 0.021$, as compared to $1.06\ \mu m$ for Tm20). Thus, when branching and terminating rates are held constant, trade-offs exist between the size and consistency of the dendritic arbors, and to some degree, dendritic complexity (*Figure 7A–C*).

In light of the spatiotemporal restrictions we observed for insulin signaling and the negative regulation by Activin signaling (*Figure 8A*), we further explored whether altering branching rate during development could mitigate the trade-offs between dendritic field sizes and consistency. We considered a simple two-stage model in which an initial activation phase is followed by an inhibition phase. In a minimal model, we found that setting an initial branching rate ($0.42\ \mu m^{-1}$) slightly higher than the fixed terminating rate ($0.4\ \mu m^{-1}$), and then turning the branching rate down ($0.36\ \mu m^{-1}$) when the maximal dendritic length reaches $100\ \mu m$, large dendritic field sizes could be generated without incurring significant variability ($11.01 \pm 2.07\ \mu m$, 18.8% CV; *Figure 7D*). If the terminating rate is raised to $0.42\ \mu m^{-1}$ at the second stage, the resulting field diameter is $10.15 \pm 2.12\ \mu m$ (20.8% CV; *Figure 7D*). Thus, large and consistent dendritic sizes may be readily achieved with two-stage dendritic growth (*Figure 8B*).

## Discussion

### Insulin/TOR/SREBP signaling regulates dendrite morphogenesis

In this study, we showed that insulin/TOR/SREBP signaling positively regulates dendritic development. Insulin-like peptide DILP2 from L5 axons signals through insulin receptors on Dm8 dendrites to promote dendritic expansion via the canonical TOR pathway and SREBP. The roles of the insulin/TOR signaling pathway in cell growth and differentiation have been very well established. However, to our knowledge, this is the first study to definitively demonstrate that insulin/TOR/SREBP signaling regulates dendritic field sizes during development. In *Xenopus* tadpoles, insulin receptor is expressed in retina and optic tectum, where it regulates synapse density and visual circuit function (*Chiu et al., 2008*). However, the source of insulin or insulin-like growth factor (IGF) and the signaling pathway have not been described in developing *Xenopus* tectum. In vitro studies in cultured cells suggest that insulin/IGF signaling can not only regulate dendrite spine morphology and neurite

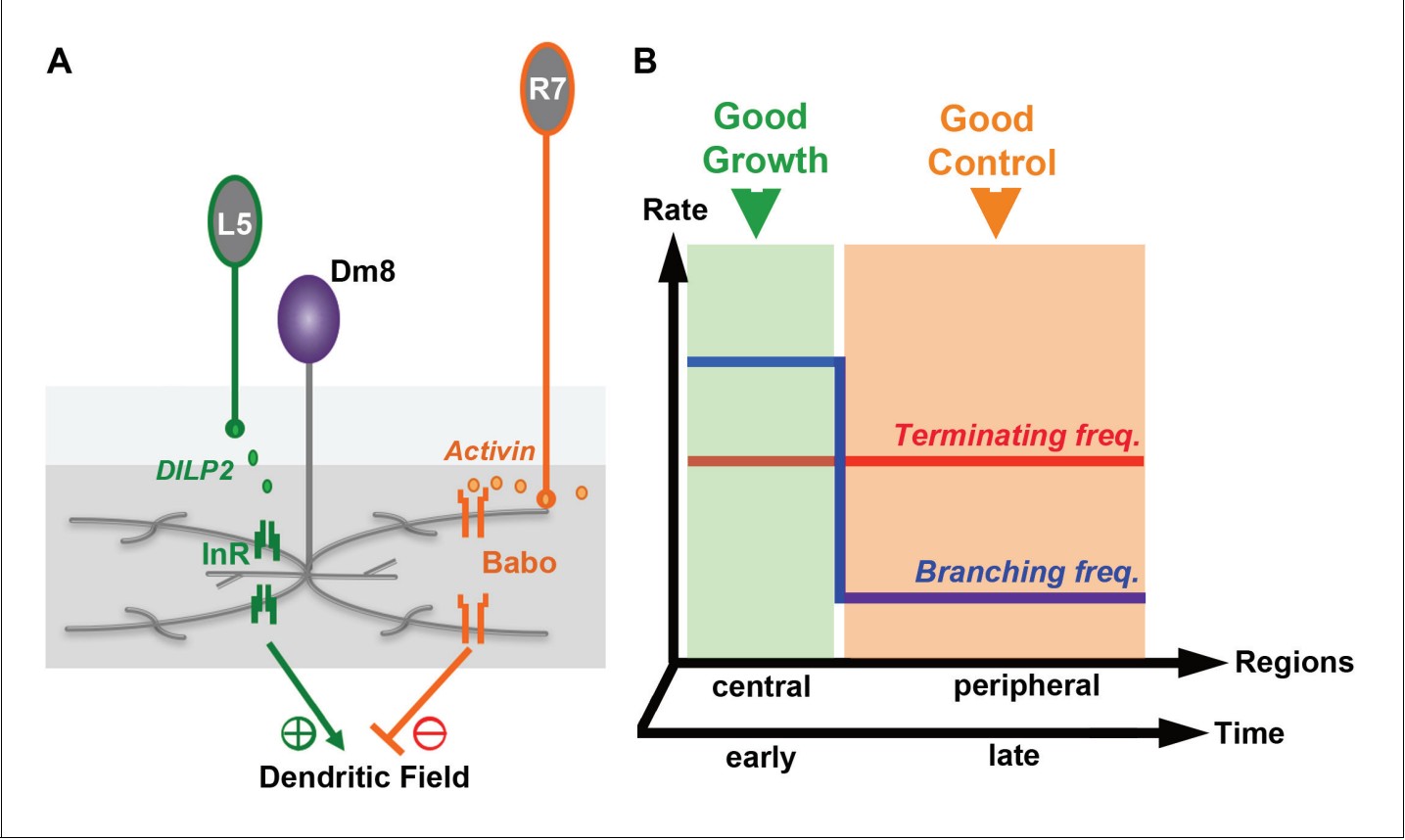

**Figure 8.** Antagonistic regulation of Dm8 dendritic field arborization by two afferent-derived factors, Activin and DILP2. (**A**) Two afferent-derived factors, Activin and insulin-like peptide2 (DILP2), signal to Dm8 amacrine neurons to control their dendritic field elaboration. Activin is provided by R7 photoreceptors, the major synaptic partners of Dm8 neurons. DILP2, on the other hand, is provided by L5 lamina neurons, which form few or no synapses with Dm8 neurons. During early developmental stages, L5-derived DILP2 locally activates InR on the Dm8 dendrites and facilitates its dendritic field expansion. In contrast, R7-derived Activin acts on Baboon in Dm8 dendrites to restrict its dendritic elaboration by increasing dendritic terminating frequency. The antagonistic regulation by DILP2 and Activin contributes to the robust and stereotyped morphology of Dm8 dendrites. (**B**) The relationship between dendritic developmental parameters (terminating and branching frequencies) and growth/robustness is shown. In the green area, the difference between the terminating and branching frequencies is small and dendritic growth is favored at the cost of robustness. Conversely, a large difference between the terminating and branching frequencies (orange area) ensures well-controlled size at the cost of overall sizes. The elaboration of large and consistent sizes of dendritic fields can be achieved by temporal and/or spatial segregation of these two phases, for example, by favoring growth in the early stages or in the center of the dendritic field.

outgrowth (*Govind et al., 2001*; *Cheng et al., 2003*), but also function as a synaptic component at the postsynaptic density (PSD) (*Abbott et al., 1999*; *Choi et al., 2005*). It would be interesting to determine whether afferent-derived insulin/IGF is also utilized for dendritic development in vertebrate nervous systems.

The PI3K/AKT/TOR signaling pathway is known to mediate a broad spectrum of functions through two major complexes, TORC1 and TORC2, which act through different and varied effectors (*Saxton and Sabatini, 2017*). In cultured hippocampal neurons, the PI3K-TOR pathway regulates dendrite size and branching patterns (*Kumar et al., 2005*; *Jaworski et al., 2005*), presumably by interacting with the microtubule plus-end-tracking protein CLIP-170 to enhance crosstalk between CLIP-170 and the actin-binding protein IQGAP1 (*Swiech et al., 2011*). Dm8 dendritic development appears to involve the TORC1 complex and one of its downstream targets, SREBP. Removing the TORC2 component, Rictor, or TORC1 targets other than SREBP had no significant effect on Dm8 dendritic field size. In contrast, SREBP mutant Dm8 neurons had a reduced dendritic field size, and overexpression of SREBP rescued the dendritic phenotype of *Tor* mutants. While we cannot rule out the possibility that other TOR targets play an additional or minor role, our results suggest that TOR works primarily through SREBP to regulate Dm8 dendritic development. Recent studies showed that

SREBP enhances the transcription of cholesterol and fatty acid synthesis enzymes to regulate dendritic development of Da neurons in the peripheral nervous system (*Meltzer et al., 2017*; *Ziegler et al., 2017*). Thus, mounting evidence suggests that lipid synthesis may be a key regulatory point for controlling dendritic development in both central and peripheral nervous systems.

Using a novel receptor-based activity-dependent GRASP (R-synGRASP) technique, we found synaptic defects between R7 photoreceptors and mutant Dm8 neurons that lack Insulin/TOR pathway signaling components. While the R-GRASP method revealed that *Tor* or *chico* mutant Dm8 dendrites formed membrane contacts with R7 axonal terminals in numbers that correlated with the dendritic field sizes, the results from the R-synGRASP method (*Figure 3*) suggested these contacts do not constitute active synapses. Notably, *Pten* mutant Dm8 neurons appear to form more active synapses with peripheral R7 photoreceptors than central R7 photoreceptors, in contrast to wild types. At this stage of inquiry, the nature of such defects remains unclear. The presynaptic structures of R7 photoreceptors appear to be unaltered, as judged by the localization of the active zone marker, Bruchpilot (*Figure 3—figure supplement 1I–J*). The level of histamine receptors on dendrites is significantly reduced in *Tor*- but not *chico* mutant Dm8 neurons, suggesting that reduced receptor levels might only partially account for the lack of R-synGRASP signal. It has been shown that insulin signaling regulates protein trafficking in various neuronal models (*Brewer et al., 2014*; *Gasparini et al., 2001*; *Kandror, 1999*), and mutations in *SREBP* and its upstream regulator *easily shocked (eas)* have been associated with excitability defects (*Pavlidis et al., 1994*; *Lim et al., 2011*; *Ziegler et al., 2017*; *Tsai et al., 2019*). Further studies will be required to resolve the nature of the synaptic defects in *Tor* and *chico* mutant Dm8 neurons.

## Antagonistic regulation of Dm8 dendritic expansion by afferent-derived factors ensures robust size control

Our previous study showed that R7-derived Activin negatively regulates the expansion of Dm8 dendritic fields (*Ting et al., 2014*). This action is countered by positive regulation by L5-derived DILP2 revealed in this study. Despite the antagonistic effects on Dm8 dendritic expansion, both Activin and DILP2 are derived from afferents, delivered to the growth cones, and likely act on dendrites at close range. Both L3 and L5 lamina neurons express DILP2 (*Tan et al., 2015*), but RNAi-mediated knockdown of *dlip2* in L5 neurons alone is sufficient to reduce Dm8 dendritic expansion. This result suggests that L3-derived DILP2 is not necessary for normal development of Dm8 dendrites, even though the axonal terminals of L3 and L5 are only few micrometers apart. Similarly, Activin derived from R7 and R8 photoreceptors acts respectively on Dm8 and Tm20 neurons, even though the R7 and R8 axonal terminals are also only separated by a few micrometers (*Ting et al., 2014*). Furthermore, both Activin and DILP2 appear to have multiple functions in the developing optic lobe. In addition to its function on Dm8 dendritic development, R7-derived Activin functions in an autocrine fashion to control R7 axonal tiling (*Ting et al., 2007*). Glia-derived DILPs regulate lamina neuron differentiation (*Fernandes et al., 2017*), while L3 and L5 neurons express DILP2, which also affects photoreceptor axonal tiling. It is thus tempting to speculate that afferent-derived factors tend to function at short range on multiple targets during distinct developmental stages.

Our results from genetic interaction experiments further argue that Activin and DILP2 signal in developing Dm8 dendrites through parallel pathways. Interestingly, the antagonistic regulation of Dm8 dendritic expansion by two afferent-derived factors is required for elaboration of appropriately large and robust Dm8 dendritic fields – removing both signaling events rendered highly variable sizes of Dm8 dendritic fields. While antagonistic regulation has been shown to control a wide range of biological processes, including gene and protein expression (*Xu et al., 2013*; *Hurtley, 2016*), to our knowledge, this work provides the first example of antagonistic regulation for robust control of dendritic field sizes (*Figure 7A*).

How does antagonistic regulation by insulin and Activin signaling ensure appropriate and robust sizes of Dm8 dendritic fields? Several lines of evidences suggest the involvement of spatiotemporal restriction of insulin signaling. Strong insulin receptor expression was detected in Dm8 neurons at early pupal stages (20–40 hr APF), when the cells had just begun to expand their dendritic arbors (*Ting et al., 2014*), but expression disappeared by late pupal stages (60 hr APF), before the dendrites expanded to adult sizes. Furthermore, the insulin receptors in developing Dm8 neurons were largely restricted to the central dendritic field, corresponding to one single column. Thus, insulin

signaling likely occurs only over a short developmental period in response to insulin signal from one or a few L5 neurons to transiently promote Dm8 dendritic expansion during the early pupal stages.

To further understand how antagonistic regulation might control dendritic field sizes, we carried out simulations of Dm8 dendritic development using a kinetic Monte Carlo method. We found that the variance of dendritic field sizes correlates with the difference between two key dendritic development parameters, branching and terminating rates: high branching and low terminating rates favor dendritic field expansion at the cost of high variability, while low branching and high terminating rates lead to small but robust dendritic field sizes (*Figure 7B*). We then found that elaboration of large and consistent dendritic fields could be achieved by spatial and/or temporal segregation of growing and terminating phases (*Figure 7D*). A previous study using in vivo imaging of dendritic dynamics showed that dendrites are elaborated at different rates according to the developmental stage (*Wu et al., 1999*). Thus, we speculate that spatiotemporal restriction of insulin signaling, coupled with Activin-mediated negative regulation, allows Dm8 neurons to elaborate a large and consistent dendritic field (*Figure 8*). The large and robust sizes of Dm8 dendritic fields not only ensure a consistent receptive field for approximately 14 R7 photoreceptor inputs (*Gao et al., 2008*), but also can receive negative feedback signals to generate a center-surround signal structure (Li and Lee, unpublished).

## Materials and methods

### Genetics

Fly stocks were maintained on standard fruit fly medium at 25°C. Transgenic flies were generated using standard P-element or PhiC31-mediated transformation protocols by Rainbow Transgenic Flies, Inc To enhance RNAi knockdown efficiency, we genetically introduced a copy of Dicer2 and conducted experiments at 28°C.

### Fly stocks

(1-5) GMR-Gal4, long-GMR-Gal4, GMR9B08-Gal4, GMR27G05-Gal4, GMR24F06-Gal4 (Bloomington Stock Center, BDSC), (6) 6–60 Gal4 (*Nern et al., 2008*), (7) Ort$^{C2b}$-Gal4 (*Gao et al., 2008*), (8) Imp-L2-RA-Gal4 (*Bader et al., 2013*), (9) DIP-γ-Gal4 (*Carrillo et al., 2015*), (10) Rh4-Brp::Short$^{mCherry}$ (*Berger-Müller et al., 2013*), (11) 24F06-LexA(BDSC), (12) Ort$^{C2b}$-LexA (*Ting et al., 2014*), (13-14) UAS-Dilp2, UAS-Dilp6 (*Ikeya et al., 2002*), (15) UAS-Dilp2::GFP (*Wong et al., 2012*), (16) UAS-Dilp2-RNAi (#102158, Vienna Stock Center), (17) UAS-Dilp6-RNAi (#102465, Vienna Stock Center), (18-19) UAS-mCD8::GFP, UAS-mCD8::mCherry (BDSC), (20) UAS-Tor$^{WT}$ (BDSC), (21) UAS-Babo$^{DA}$ (*Ting et al., 2014*), (22) UAS-Babo$^{DN}$ (*Ting et al., 2014*), (23) UAS-IVS-R::PEST (*Nern et al., 2011*), (24) UAS-SREBP$^{WT}$ (BDSC), (25)UAS-SREBP$^{CA}$ (BDSC), (26) Rh3-Syb::spGFP1-10 (this study), (27) Rh4-Syb::spGFP1-10 (this study), (28) UAS-Ort::spGFP11::HA (this study), (29) UAS-myr-TdTom::dT2A::spGFP1-10 (this study), (29) UAS-FSF-myr-TdTom::dT2A::spGFP1-10 (this study), (30) LexAop-mCD8::GFP (BDSC), (31) LexAop-FSF-mCD8::GFP (*Ting et al., 2014*), (32) LexAop-FSF-rCD2::mCherry (*Ting et al., 2014*), (33) sev$^{E2}$ (*Ting et al., 2014*), (34) hsFLP1(BDSC), (35) hsFLP122 (BDSC), (36) FRT82,InR$^{273}$ (*Song et al., 2003* Science), (37) FRT82, InR$^{353}$ (*Song et al., 2003* Science), (38) FRT40,chico$^{1}$ (*Böhni et al., 1999*), (39) FRT40,chico$^{fs(2)4}$ (*Böhni et al., 1999*), (40) FRT40,Pten$^{2L117}$ (*Oldham et al., 2002*), (41) FRT82,Tsc1$^{1A2}$ (*Stocker et al., 2003*), (42) FRT82,Tsc1$^{Q87X}$ (*Tapon et al., 2001*), (43) FRT82,Rheb$^{3M2}$, (44) FRT40,Tor$^{ΔP}$ (*Zhang et al., 2000*), (45) FRT82,foxo$^{Δ94}$ (*Slack et al., 2011*), (46) FRT2A, s6k$^{l-1}$ (*Montagne et al., 1999*), (47) FRT40,Thor$^{k07736}$ (*Spradling et al., 1999*), (48) FRT40,Dref$^{KG09294}$ (*Bellen et al., 2004*), (49) Df(3R)PI3K92E$^{A}$ (*Weinkove et al., 1999*), (50) FRT19A, rictor$^{Δ2}$ (*Hietakangas and Cohen, 2007*), (51) FRT2A, SREBP$^{189}$ (*Kunte et al., 2006*), (52) FRT42, Atg7$^{d06996}$ (Kyoto #114560), 53) FRT19A, Raptor$^{Del}$ (*Li et al., 2019*), (54-55) InR$^{E19}$, InR$^{339}$ (*Kao et al., 2015*), (56-59) InR$^{93Dj-4}$, Df(3R)ED6058, UAS-InR$^{wt}$, UAS-InR$^{K1409A}$ (BDSC).

### Molecular biology

Transgenic constructs were generated by standard subcloning, PCR, and In-Fusion cloning. All PCR amplifications were performed with Phusion High-Fidelity DNA polymerase (Thermo Scientific). The constructs were confirmed by sequencing. The cloning procedures and PCR primers are described as follows.

### *InR*::V5::spGFP11

A CRISPR target (ATAACACACCTTTCCGTAGATGG) was found on the C-terminus of *InR* using the CRISPR TARGET FINDER on FLYCRISPR (http://flycrispr.molbio.wisc.edu). The CRISPR target was sequence verified using the following primers: Forward 5′-AACGAGCAAGTCCTGCGTTA-3′ and Reverse 5′-TCCGTCCGGTTTCTTGTTCA-3′. Single-guide RNA (sgRNA) was obtained by annealing two oligos (Eurofins Genomics) and was sub-cloned into the *Bbs*I site of pU6-*Bbs*I-chiRNA (Addgene 45946, *Gratz et al., 2013*) to obtain the pU6-*Bbs*I-*InR*-chiRNA construct. Sense oligo: 5′-CTTCGA TAACACACCTTTCCGTAGA-3′ and Antisense oligo: 5′-AAACTCTACGGAAAGGTGTGTTATC-3′.

The 5′ and 3′ homologous fragments of *InR* were PCR amplified from flies bearing nos-Cas9 transgene (BDSC #36304) and sub-cloned into the 5′-*Eco*RV and 3′-*Sna*BI sites of the targeting vector (pUC57-V5- piggy-3xGFP11-GMR-3xp3-tagRed) by In-Fusion cloning. The following primers were used for amplification of 5′ and 3′ homologous fragments. For the 5′ homologous fragment of *InR*:

Fwd: 5′-CGAATGCATCTAGATATCGACTTACGTTGTGCCTGATGC-3′
Rev: 5′-GGGATAGGAGCCGATATCCGCCTCCCTTCCGATGAATC-3′

For the 3′ homologous fragment of *InR*:

Fwd1: 5′-TTCTAGGGTTAATACGTATCGTTACGAACTGTAGTTCTGTAG-3′
Rev1: 5′-GTGTCATGGATCTACGGAAAGGTGTGTTATTTTACGAAG-3′
Fwd2: 5′-CGTAGATCCATGACACCTTTTCTTCTTGGTGTCGAACCA-3′
Rev2: 5′-CCGGGATCCGATTACGTAGAAAGTGGTATTCCTAGAGCAGAGC-3′

The InR targeting plasmid (500 ng/µl) and the sgRNA (100 ng/µl) were co-injected into *Drosophila* embryos bearing the nos-Cas9 transgene (BDSC #36304, Rainbow Transgenic, CA). Flies were screened for eyes with RFP fluorescence after crossing with *yw*. The primers used for transformant validation are as follows:

M13F: 5′-GTAAAACGACGGCCAG-3′
M13R: 5′-CAGGAAACAGCTATGAC-3′
Piggy-5TR-Rev: 5′-ATGTCCTAAATGCACAGCGACG-3′
Piggy-3TR-Fw: 5′-CTGCATTCTAGTTGTGGTTTGTCC-3′

The resulting *InR*::V5::spGFP11 allele is fully viable over strong and null *InR* mutant alleles (*InR*[339], *InR*[E19], *InR*[93Dj4], and *Df(3R)ED6058*[150251]).

### Construction of pCaST-Rh3(Rh4)::Syb::spGFP1-10

Syb::spGFP1-10 fragment was PCR amplified from pUAST-Syb::spGFP1-10 (*Macpherson et al., 2015*) and used to replace the Brp::mCherry fragment in pCaST-Rh3(Rh4)::Brp::mCherry (*Ting et al., 2014*) through 5′-*Pme*I and 3′-*Xba*I sites, making pCaST-Rh3(Rh4)::Syb::spGFP1-10. The constructs were sequence verified with the following primers.
For pCaST-Rh3::Syb::spGFP1-10:

Fwd: 5′-AGGGCGAATTC<u>GTTTAAAC</u>ATGGCGGACGCTGCAC-3′
Rev: 5′-ACAAAGATCC<u>TCTAGA</u>CTATGTTCCTTTTTCATTTGGATCTTTGCT-3′

For pCaST-Rh4::Syb::spGFP1-10:

Fwd: 5′-CGGGTTGACCG<u>GTTTAAAC</u>ATGGCGGACGCTGCAC-3′
Rev: 5′-ACAAAGATCC<u>TCTAGA</u>CTATGTTCCTTTTTCATTTGGATCTTTGCT-3′

### Construction of UAS-spGFP11::HA::Ort

A synthesized DNA fragment (Eurofins Genomics) containing spGFP11 and two copies of HA was cloned into the *Msc*I site of pBS2-Ort (*Takemura et al., 2011*) to make pBS2-spGFP11::HA::Ort. The spGFP11::HA::Ort fragment was removed from pBS2-spGFP11::HA::Ort as a *Kpn*I to *Eco*RI fragment and then sub-cloned into pUAST to make UAS-spGFP11::HA::Ort. The sequence of the synthesized DNA fragment is as follows: 5′-TTCAACAAAGTCTGGCCATACGCGACCACATGGTGCTGCACGAG TACGTGAACGCCGCCGGCATCACCGGCTCCTACCCCTACGACGTGCCCGACTACGCCGGCTA TCCCTATGACGTCCCGGACTATGCAGGCTCCTTGGCCATAACCGACATCC-3′.

## Construction of UAS-myr::TdTom::dT2A::spGFP1-10 and UAS-FSF-myr::TdTom::dT2A::spGFP1-10

The myr::tdTomato, *Drosophila* codon optimized dT2A, and spGFP1-10 were PCR amplified and cloned into pUAST to make UAS-myr::TdTom::dT2A::spGFP1-10. Transcriptional terminators SV40 and αTubulin 84B with flanking FRT were inserted immediately prior to myr::tdTomato::dT2A::spGFP1-10 to generate the flippable reporter construct UAS-FSF::myr::TdTom::dT2A::spGFP1-10. The detailed strategies of molecular cloning are available upon request.

## MARCM and Flip-out

To generate the Dm8 MARCM clones, hs-Flp122 or hs-FLP1 was used with the appropriate FRT chromosomes. Three Gal4 drivers (Ort$^{C2b}$-Gal4, GMR24F06-Gal4, DIP-γ-Gal4) were used in different experiments. Late third-instar larvae were transferred to a PCR tube and subjected to heat shock in a 37°C water bath (1–3 min for hs-FLP122; 30 min for hs-FLP1). For stochastic labeling of Dm8 cells, we used pLexAop >stop > mCD8GFP and pLexAop >stop > rCD2::mCherry transgenes in combination with cell-specific drivers and hs-FLP1. To activate flipase expression, third-instar larvae bearing these transgenes were subjected to a 7 min heat shock in a 37°C water bath.

## GRASP

Receptor-based GRASP (R-GRASP) for detecting membrane contacts between R7 and Dm8 neurons was performed using a membrane-tethered spGFP$^{1-10}$ (CD4::GFP$^{sp1-10}$) expressing in all R7s and the spGFP11 and HA-tagged Ort (GFP$^{sp11}$::HA::Ort) expressed in single Dm8 clones. The native fluorescence signal of reconstituted GFP was used. Receptor-based synaptic GRASP (R-synGRASP) for detecting active synapses based on the reconstitution of syb-spGFP1-10 (*Macpherson et al., 2015*) and GFP$^{sp11}$::HA::Ort was performed in 0-12d old flies, and native fluorescence signal of GFP was used. Syb:: GFP$^{sp1-10}$ was preferentially detected by rabbit polyclonal anti-spGFP1-10 antibody. GFP$^{sp11}$::HA::Ort was detected by rat monoclonal anti-HA antibody.

## In situ hybridization

In situ hybridization was performed as described previously (*Ting et al., 2014*). DNA templates for in vitro transcription were generated by PCR amplification with the T7 (antisense) or T3 (sense) promoter sequence added to the beginning of the primer. A DIG RNA labeling kit and T3/T7 MAXIscript in vitro transcription kit (Ambion) were used to generate DIG labeled RNA probes. The full-length cDNA clone for *Dilp2* was obtained from BDSC. Two sets of primers were used for in situ hybridization:

> T3-*dilp2*-sense-Fwd: 5′-**AATTAACCCTCACTAAAGGG**CGGCTCGACCCAACTTAATC-3′
> *dilp2*-sense-Rev: 5′-CGCGCTTGTGTGGAATCAC-3′
> *dilp2*-Antisense-Fwd: 5′-CGGCTCGACCCAACTTAATC-3′
> T7-*dilp2*-Antisense-Rev: 5′-**TAATACGACTCACTATAGGG**CGCGCTTGTGTGGAATCAC-3′
> T3-*dilp2*-sense-Fwd: 5′-**AATTAACCCTCACTAAAGGG**GATCTGGACGCCCTCAATCC-3′
> *dilp2*-sense-2-Rev: 5′-CAGAGATAATCGCGTCGACC-3′
> *dilp2*-Antisense-Fwd: 5′-GATCTGGACGCCCTCAATCC-3′
> T7-*dilp2*-Antisense-Rev: 5′-**TAATACGACTCACTATAGGG**CAGAGATAATCGCGTCGACC-3′

## Immunohistochemistry

Immunohistochemistry was performed as described previously (*Ting et al., 2007*). Primary antibodies used in this study are as follows: Mouse IgG1 αGFP (1:500, DSHB), Mouse IgG2a αEyeless (1:20, DSHB), Mouse IgG2a αDachshund (1:100, DSHB), Mouse IgG1 α24B10 (1:20, DSHB), Mouse IgG2b αHiw (GH4, 1:20, DSHB), Rat αElav (7E8A10, 1:50, DSHB), Guinea pig αToy (1:100, Gift from Uwe Walldorf), Guinea pig αDichete (1:50, Gift from Claude Desplan), Rat αDrifter (1:1000, Gift from Sarah J. Certel), Guinea pig αBsh (1:600, Gift from Makoto Sato), Rabbit αGFP (1:500, Thermo Fisher Scientific), Mouse IgG2a αGFP (1:500, Thermo Fisher Scientific), Mouse IgG2a αV5 (1:500, Thermo Fisher Scientific), Rat αCD8 (1:500, Thermo Fisher Scientific), Rat αDilp2 (1:500, Gift from Pierre Leopold), Rat αRFP (1:500, Bulldog Bio), Rat αHA (1:500, Sigma-Aldrich). The secondary antibodies including goat anti-rabbit, goat anti-mouse, goat anti-rat, and goat anti-guinea pig (coupled to Alexa 488, 568, or 647) were used at a dilution of 1:500 (Thermo Fisher Scientific). Confocal images

were acquired using an upright Zeiss LSM780 or LSM880 microscope. The raw images were $512 \times 512 \times N$ voxels (N: number of Z-sections, typically ~200), 16 bits (voxel size = 0.11 μm×0.11 μm×0.2 μm), and two or three color channels. Image stacks were deconvolved using the Huygens Professional package (Scientific Volume Imaging).

### Two-electrode voltage-clamp (TEVC) recording

Synthesis and injection of cRNA, as well as TEVC recordings from *Xenopus* oocytes were performed as described (*Li et al., 2016*). Oocytes were obtained from EcoCyte (Austin, TX, USA) and were injected with 5–50 ng cRNA synthesized in vitro from linearized template cDNA. Voltage-clamp recordings were performed 2 or 3 days post injection at room temperature (25°C). Voltage and current electrodes were filled with 3.0 M KCl, and current responses were recorded at a holding potential of −60 mV.

### Statistics

For the analyses of means, unpaired Student's t-test was used. For variance analyses, Conover test and Siegel-Tukey test, which do not depend on the distribution being normal, were used to determine differences in variance (*Conover and Iman, 1981*; *Lehmann, 1988*). Tests were carried out using Mathematica (v10.4, Wolfram Research, USA).

### Simulation

We consider a 'one-stage' model where neurons grow all their dendrites at a constant speed; each dendrite grows with a constant probability of branching ($k_b$) and terminating ($k_t$). At growing length $r$, there are $n$ dendrites.

$$\boxed{n-1} \underset{k_t n}{\overset{k_b(n-1)}{\rightleftharpoons}} \boxed{n} \underset{k_t(n+1)}{\overset{k_b n}{\rightleftharpoons}} \boxed{n+1}$$

We simulated the dendrite growth stochastically with the Gillespie's algorithm (*Gillespie, 1976*; *Gillespie, 1977*). Namely, starting with $N_0$ dendrites initially, a random number between 0 and 1, $x_1$, was first obtained to determine the length they grow until the first branching or terminating event, $\Delta r$:

$$\Delta r = -\frac{1}{n(k_b + k_t)} \log(x_1).$$

A second random number (between 0 and 1), $x_2$, was used to determine either branching, where the number of dendrites increases by 1, or terminating, where $n$ decreases by 1, events according to their corresponding rates:

$$n(r+\Delta r) = \begin{cases} n(r)+1, & \text{if } x2 < kb/(kb+kt), \\ n(r)-1, & \text{otherwise.} \end{cases}$$

It is propagated until $n$ becomes zero.

To determine the resulting dendritic field sizes, we used the Brownian motion model and took the square root of the dendrite *length r* to model the *distance* (R) from the origin of growth and the 'radii' of dendritic fields (R95) were defined as the distance of 95th percentile of all the terminal points to the origin of growth.

In a typical simulation, 100 neurons were calculated, and their mean and standard deviation of their *R95* were recorded. In this case, the initial value of $n$ ($N_0$), and the branching ($k_b$) and terminating ($k_t$) rates are the three parameters in the one-stage model. To determine an appropriate number of initial dendrites, we simulated 100 neurons using different numbers (5-50) of initial dendrites and the kinetic parameters known for Tm20 neurons ($k_b$ = 0.360 $\mu m^{-1}$ and $k_t$ = 0.594 $\mu m^{-1}$; *Ting et al., 2014*). We found that the means and standard deviations of the dendritic radii are relatively insensitive to initial dendritic numbers ($N_0$), especially for those over 20 (*Figure 7—figure supplement 1*), and used $N_0$ = 30 for all subsequent simulations.

We explored the entire parameter space of $k_b$ and $k_t$ by grid scanning and determined the corresponding radii and variations of dendritic fields (R95), and dendritic density ($\mu m/\mu m^2$). Both $k_b$ and $k_b$ were set in the range of 0.01–0.99 $\mu m^{-1}$ with 49 increment of 0.02 $\mu m^{-1}$. When branching rate is

larger than the terminating rate, the dendrite number grows exponentially, leading to infinitely large dendritic fields that has no biological meaning, so these regions were excluded from analyses. The smallest $\Delta k$ (=$k_t$ $kb$) in the first scanned set was 0.02 $\mu m^{-1}$, and the largest characteristic length of dendrites, $1/\Delta k$, being 50 $\mu m$, or the largest characteristic radii being about 7 $\mu m$, which is much smaller than those of Dm8s (approximately 10–11 $\mu m$). We then scanned for the same set of $k_b$ values but with $\Delta k$ being 0.01, 0.05, 0.025 and 0.0125 $\mu m^{-1}$ and included these data in *Figure 7*.

We also developed a two-stage growing model, where the branching ($k_b$) and terminating rates ($k_t$) changes when $r$ reaches a critical value. In this case there are two sets of $k_b$ and $k_t$ values. Neurons were simulated with the first set of rates until $r$ reaches the switching value (100 μm), and then the second set of branching and terminating rates were employed. The simulation algorithm was the same as described above except the change in $k_b$ and $k_t$. All simulations were implemented in Python 3.6.5, with modules numpy, random, and matplotlib.

## Acknowledgements

We thank Drs. Edwin S Levitan, Ernst Hafen, S Lawrence Zipursky, Leslie Pick, Linda Partridge, Mary Lily, Pierre Leopold, Sarah J Certel, Ville Hietakangas, Jianzhong Yu, Hwei-Jan Hsu for providing critical reagents, Kate O'Connor-Giles and Scott Gratz for suggestions regarding CRISPR. We thank Marcus Calkins for editing and manuscript handling. PGM was supported by the Intramural Research Program of the National Institutes of Health, CIT. This work was supported by the Intramural Research Program of the NIH, the *Eunice Kennedy Shriver* National Institute of Child Health and Human Development (grant HD008913 to C-HL) and Academia Sinica, Taiwan (internal grant to C-HL). C-P H acknowledges the support from Academia Sinica Investigator Award (AS-IA-106-M01) and the Ministry of Science and Technology of Taiwan (project 105–2113 M- 001–009-MY4). This work was also benefited from events organized by the National Center for Theoretical Sciences.

## Additional information

### Funding

| Funder | Grant reference number | Author |
| --- | --- | --- |
| Academia Sinica | | Chi-Hon Lee |
| National Institutes of Health | HD008913 | Chi-Hon Lee |
| Academia Sinica | AS-IA-106-M01 | Chao-Ping Hsu |
| Ministry of Science and Technology, Taiwan | 105-2113-M- 001-009- MY4 | Chao-Ping Hsu |
| National Institutes of Health | | Philip McQueen |

The funders had no role in study design, data collection and interpretation, or the decision to submit the work for publication.

### Author contributions

Jiangnan Luo, Conceptualization, Data curation, Formal analysis, Investigation, Visualization, Methodology; Chun-Yuan Ting, Data curation, Methodology; Yan Li, Data curation, Visualization, Methodology; Philip McQueen, Formal analysis; Tzu-Yang Lin, Validation; Chao-Ping Hsu, Data curation, Software, Formal analysis, Funding acquisition, Investigation, Visualization; Chi-Hon Lee, Conceptualization, Resources, Supervision, Funding acquisition, Project administration

### Author ORCIDs

Chun-Yuan Ting (iD) https://orcid.org/0000-0002-2302-4203
Chi-Hon Lee (iD) https://orcid.org/0000-0002-6138-711X

### Decision letter and Author response

Decision letter https://doi.org/10.7554/eLife.50568.sa1
Author response https://doi.org/10.7554/eLife.50568.sa2

# Additional files

## Supplementary files
- Source code 1. Simulation programs.
- Supplementary file 1. Summary of experimental genotypes.
- Supplementary file 2. Statistics.
- Supplementary file 3. Key Resources Table.
- Transparent reporting form

## Data availability
Data generated or analyzed during this study are included in the manuscript and supporting files. The source files for computer programs have been provided.

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
