## [Decision Letter]

**Acceptance summary:**

Building on previous work, this study expands our understanding of dendrite morphogenesis and brings into focus the complex intercellular interactions that govern the assembly of the brain. The quality of the work is excellent and it is well presented. This study reports that L5-derived DILP2 locally activates transiently expressed insulin receptors in central dendrites of Dm8 neurons to positively regulate dendritic field elaboration. This positive control of dendritic growth is shown to act in parallel with the previously described negative effect of Activin signaling. The study also introduces two new transgenic tools that will be of interest to the greater community.

**Decision letter after peer review:**

Thank you for submitting your article "Antagonistic regulation by insulin and activin ensures the appropriate dendritic field sizes of amacrine neurons" for consideration by *eLife*. Your article has been reviewed by three peer reviewers, and the evaluation has been overseen by K VijayRaghavan as the Senior and Reviewing Editor. The following individual involved in review of your submission has agreed to reveal their identity: Sonia Sen (Reviewer #2).

The reviewers have discussed the reviews with one another and the Reviewing Editor has drafted this decision to help you prepare a revised submission.

Summary:

This work provides an excellent example of how multiple signaling pathways are coordinated to regulate the development of dendritic arbors of specific neurons in the *Drosophila* visual system. In this elegant study, Luo et al. describe the mechanism involved in the regulation of dendritic positioning of Dm8 amacrine neurons. The authors show that the canonical Tor signaling/InR/PI3K pathway positively regulates Dm8 dendritic field size in a cell-autonomous manner. They generate Dm8 clones for various mutant molecules present in the TOR signaling pathway and provide convincing evidence for decreased dendritic fields. They also show that SREBP, a transcriptional regulated of lipid synthesis, likely functions downstream of TOR to cell-autonomously promote Dm8 dendritic growth. Further, they show that the lamina L5 neurons that are in close proximity to R7 and Dm8 neurons, secrete DILP2 that acts on InRs that transiently express on Dm8 dendrites during earlier developmental stage. The authors further make a case for Insulin (a positive growth signal) and Activin (a negative growth signal) acting in parallel to control Dm8 dendritic development.

Overall, the findings are interesting, the paper well written, and the data well documented and presented. In the process, the authors have also generated tools that will be valuable for the community.

Essential revisions:

1) The authors demonstrate fewer membrane contacts between R7 and *Tor* mutant Dm8 neurons, and further show reduction in Dm8 Ort expression suggesting reduced synapse formation. Since the central R7 column seems to be the major synaptic partner for Dm8 neurons, are the peripheral connections important? Does reduced dendritic size in Dm8 amacrine neurons manifest reduction in pre-synaptic markers Syb and Brp, thus smaller dendrites would form fewer connection with post-synaptic projection neurons (Tm5 in the medulla)? The authors also show that there is no change in presynaptic markers Syb and Brp in R7 terminals around *Tor* mutant Dm8 clones. Could this correspond to direct connections between R7 and Tm5 (Gao et al., 2008)?

2) The authors found that a reduction in dendritic size due to loss of InR>TOR signaling resulted in a loss of synapses. There's evidence in other systems that activity-dependent homeostatic mechanisms couple dendritic fields and synapse number (e.g. Tripodi et al., 2008). It would be valuable to know what the authors' opinions are on this in the context of their findings.

3). Related to this, have the authors analyzed the Tor dendritic arborization phenotypes, in a time course in D:D cycle and L:D cycle? However, we are not suggesting this as an essential experiment, since synaptogenesis is not the focus of their manuscript.

4). In distinguishing between TORC1 and TORC2 -dependent signaling, the authors noted no significant change in dendrite size in their Rictor mutants. Did the authors test a *Raptor* mutant? And can Srebp be downstream of TOR2?

5) Since the authors have generated reagents, that will be valuable for the community at large, it will be useful to include supplementary images of the optic lobes from each individual R-GRASP and R-syn-GRASP line rather than citing data not shown.

6) Timing of the two opposing signals: The authors nicely demonstrate that InR expression in Dm8 is restricted to a narrow developmental window (40-70 APF). Is this also true for DILP2? And, importantly for activin? Their model points to a two-step process that initially maximizes growth then bring in termination to achieve the appropriate dendritic field size in a robust way. This would require Activin to come in after 70APF. Is this known? Accordingly, have the authors interfered with the timing of expression of these signals? For example, have they assessed InR overexpression phenotypes in Dm8 beyond this window or indeed when there's a premature expression of Activin?

7) In the experiments related to DILP2 RNAi, it is not clear what the WT control used was – was it also grown at 28°C? This would be important as temperature has been shown affect neurite branching. Furthermore, since Dicer has been used in their experimental conditions, have the authors done a Dicer only control at 28C.

8) In a number of experiments, the authors draw conclusions that a given manipulation did not alter dendrite morphogenesis by Dfu, though the dendrites sometimes appeared much thinner in caliber which could affect the overall dendritic volume (e.g. Figure 5N, Figure 2—figure supplement 1B, E-I). Is dendrite caliber highly variable in WT? If not, the authors could assess dendrite volume (e.g. total pixels within an ROI, or surface reconstruction). This is particularly important in Figure 2—figure supplement 1, as they use these data to narrow down their focus to SREBP.

9) Direct evidence for a genetic interaction between the two principal components of the signaling pathway under study-InR and TOR-is missing. Without such evidence, the single mutant analysis (excepting TOR and SREBP), while consistent with the pathway interpretation, is not conclusive and does not support the "InR/PI3K/TOR" short-hand for summarizing the findings. It should be possible to test this interaction in a TOR MARCM experiment (using parental lines similar to those used to generate the 2I genotype) with available 3rd chromosome resident transgenes such as UAS-RNAi against InR and/or a dominant-negative allele such as UAS-InR.K1409A.

10) The InR::V5:GFPsp11 allele is not validated as a functional allele of InR. The mechanistic interpretation of InR signaling discussed in the manuscript draws on the spatio-temporal expression of InR, as reported by this modified allele, in Dm8s. Null alleles of InR are lethal and there are a number of hypomorphic alleles of this gene available. A simple test of viability of InR::V5:GFPsp11 over a null or a strong hypomorph would be helpful in assessing the utility of this new reagent.

10b.. Primary data for InR expression in Dm8s (based on InR::V5:GFPsp11) is presented for 40 hAPF, when it is present, and 70 hAPF and adult, when it is not detectable. In the Discussion, the expression is described as follows:

"Strong insulin receptor expression was detected in Dm8 neurons at early pupal stages (20-40 h APF), when the cells had just begun to expand their dendritic arbors (Ting et al., 2014), but expression disappeared by late pupal stages (60 hAPF), before the dendrites expanded to adult sizes."

If these data, which would better define the temporal expression profile, are available, please include them in the manuscript. As detailed in Ting et al., 2014, the expansion of Dm8 dendrites takes place between 50 and 70 hAPF (Supplementary Figure 7.) Since the mechanistic interpretation of the effect of InR signaling on Dm8 dendrite growth relies on when and where InR is expressed, particularly in relation to when the dendrites have been reported to be expanding, a finer temporal characterization would be preferable.

---

## [Author Response]

Essential revisions:1) The authors demonstrate fewer membrane contacts between R7 and Tor mutant Dm8 neurons, and further show reduction in Dm8 Ort expression suggesting reduced synapse formation. Since the central R7 column seems to be the major synaptic partner for Dm8 neurons, are the peripheral connections important? Does reduced dendritic size in Dm8 amacrine neurons manifest reduction in pre-synaptic markers Syb and Brp, thus smaller dendrites would form fewer connection with post-synaptic projection neurons (Tm5 in the medulla)? The authors also show that there is no change in presynaptic markers Syb and Brp in R7 terminals around Tor mutant Dm8 clones. Could this correspond to direct connections between R7 and Tm5 (Gao et al., 2008)?

Two lines of evidence indicate that the connections between Dm8 and peripheral R7s are important. First, the activity-dependent GRASP (Figure 3) showed that these synapses are active. Second, our receptive field mapping using 2D white noise analyses revealed that the functional receptive field for single Dm8s is larger than that of single R7s (Yan Li and C.-H. Lee, unpublished), indicating that the peripheral R7s expand the receptive field of Dm8.

The reviewers further inquired whether the mutant Dm8s of reduced dendritic fields might form fewer synapses with projection neurons (such as Tm5c). Because SREBP has been showed to be critical for maintaining presynaptic vesicle pools in photoreceptors (Tsai et al., 2019), we consider this a possibility but might not be a direct consequence of reduced dendritic field sizes (because Dm8’s presynaptic sites are mostly centrally located). However, addressing this question experimentally appears to be outside the scope of our current study which focuses on dendritic development.

The reviewers also inquired whether the presynaptic sites of R7s (marked by Syb and Brp) might correspond to the synapses between R7 and Tm5. Indeed, while Dm8 is R7’s major synaptic partner, R7 is also presynaptic to other medulla neurons (Tm5, for example). So at least some of these presynaptic sites could correspond to R7-Tm5 synapses. Based on the available data, we could only conclude that R7’s presynaptic sites did not drastically change by corresponding *Tor* mutant Dm8s. We have clarified this in the text (subsection “Disrupted insulin/Tor signaling restricts dendritic fields and alters synapses”, last paragraph).

2) The authors found that a reduction in dendritic size due to loss of InR>TOR signaling resulted in a loss of synapses. There's evidence in other systems that activity-dependent homeostatic mechanisms couple dendritic fields and synapse number (e.g. Tripodi et al., 2008). It would be valuable to know what the authors' opinions are on this in the context of their findings.

We appreciate the reviewers’ pointing out homeostasis as a potential mechanism. However, we have tried several activity manipulation methods, including changing light conditions and found none changed Dm8 dendritic field sizes. Indeed, Dm8’s dendritic arborization completed by 70% APF when photoreceptors only began to respond to light. While it is difficult to exhaust all possible ways to manipulate activity, our available data suggest that Dm8’s dendritic elaboration is developmentally hard-wired.

3) Related to this, have the authors analyzed the Tor dendritic arborization phenotypes, in a time course in D:D cycle and L:D cycle? However, we are not suggesting this as an essential experiment, since synaptogenesis is not the focus of their manuscript.

We have not examined *Tor* dendritic arborization phenotypes in D:D or L:D cycle for the reasons explained above.

4) In distinguishing between TORC1 and TORC2 -dependent signaling, the authors noted no significant change in dendrite size in their Rictor mutants. Did the authors test a Raptor mutant? And can Srebp be downstream of TOR2?

All previous studies known to us place SREBP downstream of TORC1. As suggested by the reviewers, we have tested a *Raptor* null mutant and found that it phenocopies *Srebp*. The results are now described in the text (subsection “SREBP is required for Dm8 dendritic elaboration”, first paragraph; Figures 2K, 2N).

5) Since the authors have generated reagents, that will be valuable for the community at large, it will be useful to include supplementary images of the optic lobes from each individual R-GRASP and R-syn-GRASP line rather than citing data not shown.

As suggested, we now include these in Figure 3—figure supplement 1B and 1C.

6) Timing of the two opposing signals: The authors nicely demonstrate that InR expression in Dm8 is restricted to a narrow developmental window (40-70 APF). Is this also true for DILP2? And, importantly for activin? Their model points to a two-step process that initially maximizes growth then bring in termination to achieve the appropriate dendritic field size in a robust way. This would require Activin to come in after 70APF. Is this known? Accordingly, have the authors interfered with the timing of expression of these signals? For example, have they assessed InR overexpression phenotypes in Dm8 beyond this window or indeed when there's a premature expression of Activin?

The reviewers inquired about the timing of DILP2 and Activin signals. Our in situ hybridization data showed strong Activin expression (blue) in R7s at 50% APF (Author response image 1). Activin expression in R-cells beyond 50% AFP has been reported in Zhang et al., Cell report, 2016 and showed that its expression extends through development (Author response image 2). For DILP2, we carried out QPCR with lamina tissue and found that DILP2 expression dipped at 60% APF but came up in 72

% APF (Author response image 3).

**Author response image 1. respfig1:** In situ hybridization of Activin of developing eye discs..

**Author response image 2. respfig2:** Activin mRNA levels in developing R-cells (Zhang et al., 2016)..

**Author response image 3. respfig3:** Dilp2 mRNA levels in developing lamina..

As suggested, we expressed InR in Dm8 to interfere with its expression timing. Consistent with the model, perturbing the temporal expression of InR significantly increased the variability of dendritic field sizes. These new results were described in the last paragraph of the subsection “Insulin and Activin act in parallel pathways to control Dm8 dendrite development”, and Figure 6.

7) In the experiments related to DILP2 RNAi, it is not clear what the WT control used was – was it also grown at 28C? This would be important as temperature has been shown affect neurite branching. Furthermore, since Dicer has been used in their experimental conditions, have the authors done a Dicer only control at 28C.

The control (mislabeled as *wt*), which carried an UAS-Dicer transgene, was grown at 28^o^C and showed no significantly different dendritic field sizes from the wild-type. It is now correctly labeled as control in Figure 5.

8) In a number of experiments, the authors draw conclusions that a given manipulation did not alter dendrite morphogenesis by Dfu, though the dendrites sometimes appeared much thinner in caliber which could affect the overall dendritic volume (e.g. Figure 5N, Figure 2—figure supplement 1B, E-I). Is dendrite caliber highly variable in WT? If not, the authors could assess dendrite volume (e.g. total pixels within an ROI, or surface reconstruction). This is particularly important in Figure 2—figure supplement 1, as they use these data to narrow down their focus to SREBP.

Indeed, the dendrite caliber varies within in a single Dm8 as well as among different Dm8s. From serial EM reconstruction, we know that some of the dendrites are beyond the diffraction limit of light microscopy. Thus, dendrite volume as assessed by the pixels would be rather inaccurate. We now state that dendrite caliber might be altered in the mutant Dm8s but we are not able to assess the changes (subsection “Canonical TOR signaling positively regulates Dm8 dendritic field size”, first paragraph).

9) Direct evidence for a genetic interaction between the two principal components of the signaling pathway under study-InR and TOR-is missing. Without such evidence, the single mutant analysis (excepting TOR and SREBP), while consistent with the pathway interpretation, is not conclusive and does not support the "InR/PI3K/TOR" short-hand for summarizing the findings. It should be possible to test this interaction in a TOR MARCM experiment (using parental lines similar to those used to generate the 2I genotype) with available 3rd chromosome resident transgenes such as UAS-RNAi against InR and/or a dominant-negative allele such as UAS-InR.K1409A.

Indeed, the dendrite caliber varies within in a single Dm8 as well as among different Dm8s. From serial EM reconstruction, we know that some of the dendrites are beyond the diffraction limit of light microscopy. Thus, dendrite volume as assessed by the pixels would be rather inaccurate. We now state that dendrite caliber might be altered in the mutant Dm8s but we are not able to assess the changes (subsection “Canonical TOR signaling positively regulates Dm8 dendritic field size”, first paragraph).

10) The InR::V5:GFPsp11 allele is not validated as a functional allele of InR. The mechanistic interpretation of InR signaling discussed in the manuscript draws on the spatio-temporal expression of InR, as reported by this modified allele, in Dm8s. Null alleles of InR are lethal and there are a number of hypomorphic alleles of this gene available. A simple test of viability of InR::V5:GFPsp11 over a null or a strong hypomorph would be helpful in assessing the utility of this new reagent.

As suggested, we carried out complementation tests of the *InR::V5::GFPsp11* allele and find that it is fully viable over a number of *InR* null and strong hypomorphic alleles (InR^339^; InR^E19^; InR^93Dj4^; Df^150251^) (described in subsection “Molecular Biology”).

10b.. Primary data for InR expression in Dm8s (based on InR::V5:GFPsp11) is presented for 40 hAPF, when it is present, and 70 hAPF and adult, when it is not detectable. In the Discussion, the expression is described as follows:"Strong insulin receptor expression was detected in Dm8 neurons at early pupal stages (20-40 h APF), when the cells had just begun to expand their dendritic arbors (Ting et al., 2014), but expression disappeared by late pupal stages (60 hAPF), before the dendrites expanded to adult sizes."If these data, which would better define the temporal expression profile, are available, please include them in the manuscript. As detailed in Ting et al., 2014, the expansion of Dm8 dendrites takes place between 50 and 70 hAPF (Supplementary Figure 7.) Since the mechanistic interpretation of the effect of InR signaling on Dm8 dendrite growth relies on when and where InR is expressed, particularly in relation to when the dendrites have been reported to be expanding, a finer temporal characterization would be preferable.

As suggested, we now provided Dm8’s InR expression data at 50% and 60% APF. As showed in Figure 4, InR was weakly detectable at Dm8’s dendrites at 50% APF and completely disappeared at 60% APF. The results are consistent with our conclusion of spatiotemporal restriction of InR expression on Dm8 dendrites. These results are now described in the last paragraph of the subsection “Endogenous insulin receptors are localized to Dm8 dendrites”.